# The nature of non-phononic excitations in disordered systems

Walter Schirmacher [1,2] ✉, Matteo Paoluzzi [3,4,5], Felix Cosmin Mocanu [6,7], Dmytro Khomenko [5], Grzegorz Szamel [8], Francesco Zamponi [5,7] & Giancarlo Ruocco [2,5] ✉

The frequency scaling exponent of low-frequency excitations in microscopically small glasses, which do not allow for the existence of waves (phonons), has been in the focus of the recent literature. The density of states $g(\omega)$ of these modes obeys an $\omega^s$ scaling, where the exponent $s$, ranging between 2 and 5, depends on the quenching protocol. The orgin of these findings remains controversal. Here we show, using heterogeneous-elasticity theory, that in a marginally-stable glass sample $g(\omega)$ follows a Debye-like scaling ($s = 2$), and the associated excitations (type-I) are of random-matrix type. Further, using a generalisation of the theory, we demonstrate that in more stable samples, other, (type-II) excitations prevail, which are non-irrotational oscillations, associated with local frozen-in stresses. The corresponding frequency scaling exponent $s$ is governed by the statistics of small values of the stresses and, therefore, depends on the details of the interaction potential.

Understanding the nature of vibrational states in glasses is crucial for gaining insight into their mechanical and thermal properties[1]. Correspondingly a large amount of experimental[1–10] theoretical[11–22] and simulational work[23–30] has been undergone in the last ~50 years. A paradigm for the anomalous vibrational features of glasses is the so-called boson peak (BP), which is an enhancement of the vibrational density of states (DOS) $g(\omega)$ over Debye's $g(\omega) \propto \omega^2$ law, where $\omega = 2\pi\nu$ is the angular frequency, and $\nu$ is the frequency. Such an enhancement is observed in experimental studies of macroscopically large samples. The nature of the boson-peak anomaly has been debated controversially[31,32] (see[33] for a listing and a discussion of various proposed models for the the the boson peak), but – as we feel – in the light of heterogeneous-elasticity theory (HET)[19,20,34] it became clear that the boson peak marks a cross-over of the vibrational state's nature from occasionally scattered plane waves ("phonons") at low frequencies, as considered by Debye[35], to random-matrix type states in the BP region, as demonstrated first by

Schirmacher et al.[14]. To avoid confusion, from now on we will call these random-matrix type states 'type-I non-phononic' states.

On the other hand, in simulations of samples of reduced size, a different type of low-frequency modes was detected. We call these modes "type-II" non-phononic modes.

As early as 1991 Laird and Schober[23] found, in a simulation of a glass made up of 500 particles, low-frequency states that appeared to be localized, as estimated from the participation ratio. They coined the term "quasi-localized" excitations, because in a larger system these excitations must hybridize with the waves, like local oscillatory defects in crystals[36]. Such non-phononic modes, which become visible at low frequency in small samples have more recently attracted a lot of attention[37,38,38–41]. Many of the reported type-II non-phononic excitations have been found to exhibit a DOS $g(\omega) \propto \omega^s$ scaling with $s = 4$. Recently, in a detailed MD study of low-frequency non-phononic excitations, a continuous change of the DOS exponent from $s = 4$ to the Debye-like

[1]Institut für Physik, Staudinger Weg 7, Universität Mainz, D-55099 Mainz, Germany. [2]Center for Life Nano Science @Sapienza, Istituto Italiano di Tecnologia, 291 Viale Regina Elena, I-00161 Roma, Italy. [3]Istituto per le Applicazioni del Calcolo del Consiglio Nazionale delle Ricerche, Via Pietro Castellino 111, 80131 Napoli, NA, Italy. [4]Departament de Física de la Matèria Condensada, Universitat de Barcelona, Carrer de Martí i Franquès 1, 08028 Barcelona, Spain. [5]Dipartimento di Fisica, Universita' di Roma "La Sapienza", P'le Aldo Moro 5, I-00185 Roma, Italy. [6]Dept. of Materials, Univ. of Oxford, Parks Road, Oxford OX13PH, UK. [7]Laboratoire de Physique de l'Ecole Normale Supérieure, ENS, Université PSL, CNRS, Sorbonne Université, Université Paris-Diderot, Sorbonne Paris Cité, Paris, France. [8]Dept. of Chemistry, Colorado State University, Fort Collins, CO 80523, USA. ✉e-mail: walter.schirmacher@uni-mainz.de; giancarlo.ruocco@iit.it

value $s = 2$ has been reported, depending on the quenching protocol[42–44]. In one approach[42], a fraction of the particles was fixed in space during the quenching process (pinned particles). With increasing fraction of pinned particles $s$ was found to increase continuously from 2 to 4 (and even above). In another approach[43,44], the glass at $T=0$ was produced by quenching from a well equilibrated liquid temperature $T^*$ (the "parental temperature"). Upon increasing $T^*$ from the (numerical) dynamical arrest temperature $T_d$, the exponent of the DOS in the resulting glass was reported to decrease continuously from 4 to 2, up to $T^*$ roughly twice the value of $T_d$, and to remain Debye-like for higher parental temperatures. It is worth to emphasise that the Debye-like DOS with $s = 2$ found in[43,44] has no relation with wave-like excitations, which, in the simulated small systems, can only exist at high enough frequencies.

As pointed out by Paoluzzi et al.[44], the observed $s = 2$ may be related to the vicinity of a marginal stability transition[22,45,46]. At this transition, unstable modes would appear if the glass temperature would be slightly increased. In fact, it has been demonstrated within mean-field spin glass theory[22,47,48] that in the case of marginal stability the DOS obeys an $\omega^2$ law, which is due to a Gaussian-Orthogonal Ensemble (GOE) type or Marchenko-Pastur-type random-matrix statistics (and not due to waves). In models with spatially fluctuating force constants[14,16,18,49] and in the HET model (spatially fluctuating elastic constants)[19,20,28,50], marginal stability appears if the amount of negative force or elastic constants approaches a critical threshold value. If this amount is increased beyond this threshold, again, unstable modes appear in the spectrum (with negative eigenvalues). Such unstable spectra, calculated with HET theory, in fact, have been recently used to model the instantaneous spectrum of liquids[51].

Given the observed scenario, one can argue that quenching from a high parental temperature $T^*$ the system may reach a situation of marginal stability, in which the type-I non-phononic excitations extend to zero frequency and $s$ takes the value of 2, as predicted by mean field theory and HET for the case of marginal stability. Quenching from a lower $T^*$ might enable the liquid to accomodate in a more comfortable situation, in which the type-I modes are confined to the region above a finite frequency, which would be the BP frequency in macroscopically large systems. In small samples, which do not allow for the existence of waves at small frequency, there is now room for the appearance of type-II non-phononic modes. Evaluating the spectrum of these type-II modes, and explaining their origin, is the main scope of the present paper.

Before characterizing the type-II non-phononic excitations, we show that HET predicts a DOS $g(\omega) \sim \omega^2$ at the marginally stable limit. We then derive a generalization of heterogeneous elasticity theory, by using the continuum limit of the system's Hessian. We unveil the nature of the type-II non-phononic excitations, being irrotational, vortex-like displacement fields, associated with local, frozen-in stresses.

We find a direct relationship between the statistics of the local stresses, governed by small values of the first derivative of the interaction potential, and the DOS of the type-II non-phononic modes. From this relationship it follows that the value $s = 4$ often found in numerical simulation[41] is the consequence of the cubic smoothing (tapering) of the potential at its cutoff. We tested this with numerical simulations, demonstrating that the value of $s$ is changed if the tapering function is altered. A further consequence of the relation between the internal stresses and the spectrum of the type-II spectrum is a scaling of $s = 5$ for potentials with a minimum, such as the Lennard-Jones (LJ) potential. This scaling is expected to be generic for systems with both attractive and repulsive interactions, and has been observed recently in simulations of small disordered LJ systems[52,53].

## Results

### Type-I nonphononic modes and Heterogeneous-Elasticity Theory (HET)

Heterogeneous-elasticity theory (a derivation from the microscopic Hessian and a brief description of the main steps of the theory are given in paragraph M1 of the Methods section) is based on the assumption of a spatially fluctuating local shear modulus $G(\mathbf{r}) = G_0 + \Delta G(\mathbf{r})$. Here $G_0$ is the average of the shear modulus, and the fluctuations $\Delta G(\mathbf{r})$ are supposed to be short-range correlated (see Methods M1). The bulk modulus $K$ is supposed to be uniform. Such fluctuations can be derived by a coarse-graining procedure (see[54,55] and Methods M1) from the Hessian matrix of the glass, i.e. the second-order, harmonic Taylor coefficients of the total energy. The statistics of the fluctuations has been verified in a simulation of a soft-sphere glass[28]. Using a mean-field theory derived by field-theoretical techniques (self-consistent Born approximation, SCBA), the mesoscopic spatially fluctuating part of the shear modulus is transformed into a complex, frequency dependent self energy $\Sigma(z)$, which is the central quantity for the discussion of the influence of the disorder on the spectrum. Here $z = \lambda + i0 \equiv \omega^2 + i0$ is the spectral parameter. The imaginary part $\Sigma''(\lambda)$ is proportional to $\Gamma(\omega)/\omega$, where $\Gamma(\omega)$ is the sound attenuation coefficient. It is worth to clarify once more that the HET is fully harmonic, the system dynamics is described by the Hessian, no anharmonic terms are present and the low-frequency sound attenuation is due to Rayleigh scattering of waves from the disordered structure.

It has been shown in[20] that at low frequencies $\Sigma''(\lambda)$ just adds to the spectrum $\rho(\lambda) \equiv g(\omega)/2\omega$ (where $\lambda = \omega^2$ is understood) that in absence of the self energy would be the Debye phonon spectrum. So by definition $\Sigma''(\lambda)$ describes essentially the non-phononic part of the vibrational spectrum.

The disorder-induced frequency dependence of $\Sigma(z)$ is governed by a dimensionless disorder parameter, $\gamma$, which is proportional to the variance of the shear modulus fluctuations:

$$\gamma = A \frac{\langle (\Delta G)^2 \rangle}{G_0^2}, \tag{1}$$

where $A$ is a dimensionless factor of order unity. Within the mean-field HET, on changing $\gamma$, there is a sharp crossover from stability (no negative eigenvalues $\lambda$) to instability (presence of negative $\lambda$). Near the instability, which takes place at the critical value $\gamma_c$, the self energy can be represented as[50]:

$$\Sigma(z) = \Sigma_c \left\{ 1 + \frac{2}{\gamma_c} \sqrt{\gamma_c - \gamma[1 + z\mathcal{G}(z)]} \right\} \tag{2}$$

where $\Sigma_c$ is $\Sigma(z = 0)$ at criticality and $\mathcal{G}(z)$ is a linear combination of the longitudinal and transverse Green's function (see[50] and Methods M1). Near $\lambda = 0$, $\mathcal{G}(z)$ can be represented as

$$\mathcal{G}(z) = \mathcal{G}_0 + i\mathcal{G}_1 z^{(d-2)/2}, \tag{3}$$

where the imaginary part $\mathcal{G}''$ gives rise the to the Debye spectrum:

$$\begin{aligned} \rho_\varepsilon(\lambda = \omega^2) &= \frac{1}{2\omega} g_\varepsilon(\omega) \sim \mathcal{G}''(\lambda) \\ &\sim \lambda^{(d-2)/2} = \omega^{d-2}, \end{aligned} \tag{4}$$

which is the Debye law

$$g_\varepsilon(\omega) \sim \omega^{d-1}. \tag{5}$$

The subscript $\varepsilon$ of the spectral function $\rho_\varepsilon(\lambda)$ and the DOS $g_\varepsilon(\omega)$ indicates that these excitations correspond to the HET theory, which is formulated in terms of the strain tensor $\overset{\leftrightarrow}{\varepsilon}$.

For small frequencies, $\Sigma''(\lambda)$ can be expanded with respect to $\mathcal{G}''(\lambda)$, leading to Rayleigh scattering $\Sigma''(\lambda) \sim \lambda^{d/2}$[20,21,56]. For larger frequencies $(\lambda > \lambda_c \sim \gamma_c - \gamma)$ the square-root in Eq. (2) produces its own imaginary part, leading to a shoulder in the spectrum. This shoulder appears as a maximum in the "reduced DOS" $g(\omega)/\omega^2$ and has been called 'boson

peak' for historical reasons[34]. So the boson peak is the crossover from a Debye-spectrum to a random-matrix spectrum.

If we deal with finite-size samples ($N$ particles), phonons do not exist below the first resonance frequency $\lambda_o = \omega_o^2 \sim N^{-2/d}$. In the absence of phonons, $\mathcal{G}_0$ has still a finite value, but the low-frequency imaginary part $\mathcal{G}''(\lambda)$ is gone. Instead of a boson peak Eq. (2) predicts now a *gap* in the spectrum. Beyond the gap edge (situated at $\lambda_c = \omega_c^2$) $\Sigma'(\lambda)$ increases as $\sqrt{\lambda - \lambda_c}$. This feature is shared by the mean-field theory of spin glasses[22,47,48], associated with a Marchenko-Pastur or GOE-type random-matrix spectrum. As stated above, we call these disorder-dominated random-matrix modes "type-I non-phononic excitations". The eigenvalues of these excitations obey random-matrix statistics of the GOE type, as will be demonstrated below.

The type-I modes obviously dominate the low-frequency spectrum of a marginally stable system with $\lambda_c \sim \gamma_c - \gamma = 0$, leading to $\rho_\varepsilon(\lambda) \sim \lambda^{1/2}$ for $\lambda \rightarrow 0$. Because this law (which converts to a $g_\varepsilon(\omega) \sim \omega^2$ law) is observed for quenching from a high parental temperature $T'$, we conjecture that for such high parental temperatures a marginally stable system is produced by the quenching procedure. This is rather plausible, because the quenching procedure forces the final glass to have only positive eigenvalues of the Hessian, but still keeps the original liquid structure almost unaltered. Thus, the obtained glass inherent structure lies "high" in the Potential Energy Landscape (PEL). This is different in the case of quenching from a low parental temperature, where the initial liquid structure is equilibrated into a lower energy region of the PEL. The distinction between low and high parental temperature, which is fundamental for understanding the properties of simulated glasses[42,43], is irrelevant in the case of real glasses: the quenching of a real system is anyway appreciably slower (by many orders of magnitude) than in the case of simulated glasses, which means that during the quenching the system continuously equilibrates at lower and lower temperature. Thus, except for the case of more complex disordered materials like rubbers, gels, foams or granular materials[57], an atomic or molecular glass is always somewhat away from marginality.

## Type-II non-phononic excitations and generalized heterogeneous-elasticity theory (GHET)

We now turn to the type-II non-phononic excitations, which we conjecture to appear in small samples (no low-frequency phonons) in a more stable situation $\gamma_c - \gamma \gg 0$ reached by quenching from low parental temperatures (no type-I non-phononic excitations at low frequency). In this case, almost all the numerical simulations, independently of the specific interaction potentials, indicate $s = 4$[41], although for Lennard-Jones-type potentials the situation is less clear[40,52,53,58–60]. To clarify the origin and the nature of these type-II excitations, which are predicted neither by the HET as described before, nor by the standard mean-field spin glass theory (see, however, Bouchbinder et al.[61]), we have developed a generalized version of the HET (let us call it Generalized HET, i.e. GHET) where the displacement field is described by two, instead than by a single, variables. These variables are derived in terms of a modified continuum description, in which, in addition to the usual strain $\overleftrightarrow{\varepsilon}(r,t)$ tensor field[54,62], a second−non-irrotational−vector field $\boldsymbol{\eta}(r, t)$ (vorticity) is necessary for a correct description of the dynamics. The two fields are coupled. While the features of the strains are basically controlled by the space dependence of the fluctuating elastic constants and give rise to the type-I non-phononic modes (predicted by standard HET), the vorticities are associated with spatially fluctuating local stresses $\tilde{\sigma}_{ij}^{\alpha\beta}$, which can be written as (see Methods M2)

$$\sigma_{ij}^{\alpha\beta} = \frac{1}{\Omega_Z} r_{ij}^\alpha r_{ij}^\beta \phi'(r_{ij})/r_{ij} = \frac{1}{\Omega_Z} \hat{e}_{ij}^\alpha \hat{e}_{ij}^\beta r_{ij} \phi'(r_{ij}) \qquad (6)$$

Here $\phi'(r)$ is the first derivative of the pair potential, $\boldsymbol{r}_{ij} = \boldsymbol{r}_i - \boldsymbol{r}_j$ is the vector between the positions $\boldsymbol{r}_i$ of a pair of interacting particles,

$\hat{\boldsymbol{e}}_{ij} = \boldsymbol{r}_{ij}/r_{ij}$ the corresponding unit vector, $r_{ij} = |\boldsymbol{r}_{ij}|$ is the interparticle distance, and $\Omega_Z$ is a small volume around the center-of gravity vector $\boldsymbol{R}_{ij} = \frac{1}{2}[\boldsymbol{r}_i + \boldsymbol{r}_j]$

As shown in the Methods M2 section, we are able to demonstrate that the dynamics of $\boldsymbol{\eta}(r, t)$ is similar to that of a set of local oscillators, coupled to the strain field. To the best of our knowledge, such vortex-like harmonic vibrational excitations have not yet been considered.

It has been pointed out by Alexander[57,63] that the term in the potential energy, which involves the local stresses, violates local rotation invariance. In fact, the presence of frozen-in local stresses is the reason for the existence of the type-II excitations. The local stresses also provide the coupling between the type-II excitations and the strain field.

In order to be more specific, but, as well, avoid the introduction of too complicated technicalities, we use a simplified version of GHET (Methods M2), where the non-irrotational fields and the local stresses are treated as scalars. The additional contribution to the self energy is therefore (see Methods M2)

$$\Sigma_{\varepsilon\eta}(z) \sim \overline{\frac{(\sigma_{ij}^{\alpha\beta})^2}{-z\zeta + \sigma_{ij}^{\alpha\beta}}} = \int d\sigma \mathcal{P}(\sigma) \frac{\sigma^2}{-z\zeta + \sigma}. \qquad (7)$$

Here $\sigma$ (without indices) denotes any component of the stress tensor $\sigma_{ij}^{\alpha\beta}$, and $\zeta$ is a local moment-of-inertia density (see Methods M2). The overbar indicates the average over the local stresses, and $\mathcal{P}(\sigma)$ is their distribution density. The local stresses may assume both positive and negative values. Because the local stresses include those due to the image particles of the periodic boundary conditions, even for a strictly repulsive potential the diagonal elements $\sigma^{\alpha\alpha}$ may be positive and negative, see Methods M2 and §17 of[57]. Only positive values of $\sigma$ contribute to the spectrum.

The corresponding contribution to the spectrum $\rho_{\varepsilon\eta}(\lambda)$ is

$$\rho_{\varepsilon\eta}(\lambda) \sim \Sigma_{\varepsilon\eta}''(\lambda) \sim \sigma^2 \mathcal{P}(\sigma)\big|_{\sigma = \lambda\zeta}. \qquad (8)$$

Equation (8) represents one of the main results of the present paper. Once the distribution of the local stresses is known, we have an explicit expression for the contribution of the type-II non-phononic modes to the DOS. If, for example, $\mathcal{P}(\sigma) \sim \sigma^{-1/2}$, from Eq. (8) one obtains $\rho_{\varepsilon\eta}(\lambda) \sim \lambda^{3/2}$, which leads to $g_{\varepsilon\eta}(\omega) \sim \omega^4$.

The distribution of the quantity $\sigma_{ij}^{\alpha\beta}$ can be easily derived (Methods M3) assuming an isotropic system, and from the knowledge of the particles' radial pair distribution function $g_2(r)$. Indeed, the stress distribution $\mathcal{P}(\sigma)$ is related to the particle distance distribution via (Methods, M3)

$$\mathcal{P}(\sigma) = r^{d-1} g_2(r) \left|\frac{d\sigma}{dr}\right|^{-1}, \qquad (9)$$

where $\sigma(r) = r\phi'(r)/2\Omega_c$.

How to rationalise the (almost) always observed $s = 4$? In the numerical simulations, in order to speed up the calculations and to optimally deal with the periodic boundary conditions, a cut-off radius is introduced, i.e. the interaction potentials is forced to be zero for distances larger than a given interparticle distance $r_c$. Furthermore, to guarantee the stability of the dynamics, and the numerical conservation of the total energy, the interaction potential is adjusted − using a proper tapering function − in such a way to have at least the first two derivatives continuous at $r_c$. Besides details specific to each simulation, the general form of the *employed* interaction potential is:

$$\phi_{DM}(r) = \left(\phi(r) - \phi(r_c)\right) T_m(r/r_c), \qquad (10)$$

where $T_m(x)$ ($m \geq 2$) is the tapering function, which is zero for $x \geq 1$, and whose first $m$ derivatives are also zero at $x = 1$. This guarantees that the

first $m$ derivatives of the potential are continuous at $r = r_c$. In other words, the interaction potentials employed in the numerical simulations always vanish at $r_c$ with a power law $(r_c - r)^{(m+1)}$. In most simulations the value $m = 2$ is taken, which is the minimum value to guarantee that the first two derivatives of the potential are continuous at $r = r_c$.

Because the DOS of the type-II non-phononic modes is highly sensitive to the shape of the interaction potentials close to the zeros of its first derivative, one can expect that the tapering function has a large effect on the DOS. Indeed (Methods M3), we find that, for any inverse-power law potential, in presence of a tapering function one has

$$\mathcal{P}(\sigma) \sim \sigma^{-1+\frac{1}{m}}, \tag{11}$$

and consequently

$$
\begin{aligned}
\rho_{\varepsilon\eta}(\lambda) &\sim \lambda^{1+\frac{1}{m}} \\
g_{\varepsilon\eta}(\omega) &\sim \omega^{3+\frac{2}{m}}.
\end{aligned} \tag{12}
$$

Thus, in this case, the DOS of the type-II non-phononic modes is controlled by the tapering function, not by the details of the glass or of the interaction potentials. The (almost) universally used $m = 2$ gives for the type-II low-frequency DOS the scaling $g_{\varepsilon\eta}(\omega) \sim \omega^4$, i.e. the $s = 4$ scaling of the low-frequency DOS, which has been observed in many recent small-sample simulations[41].

In the case of a potential with a smooth, tapered cutoff, the scale of the relevant values of $\sigma$ is determined, among other constants, by $g_2(r_c)$. The latter quantity actually depends on the thermodynamic state of the system and on the quenching protocol, and this observation could explain the findings that the intensity of the type-II non-phononic mode spectrum does actually depend on the glass structure: see Fig. 2 in Ref. 64 and Fig. 10 in Ref. 60.

A final consideration concerns the rather interesting case of potentials with repulsive *and* attractive parts like the frequently used Lennard-Jones (LJ) potential. In this case the first derivative of the potential – relevant to the stresses—vanishes linearly at the potential minimum. Because $g_2(r)$ has a maximum there, the corresponding small stresses will dominate over the tapering-induced ones. At the potential minimum the stress goes to zero, but the second derivative does not, which corresponds to the $m = 1$ tapering, leading to a DOS $g_{\varepsilon\eta}(\omega) \sim \omega^5$. In fact, such a scaling has been recently observed in simulations of small LJ systems[52,53]. These authors thus conjectured that the $s = 5$ scaling is generic to LJ systems. In our view this is true and not only applies to the LJ case but also to all potentials with attractive and repulsive contributions.

In other simulations of small LJ systems the spectrum does not clearly show a simple power law[58–60]. Here we may assume that in addition to the small stresses due to the potential minimum, contributions from the smooth cutoff and/or type-I excitations may be involved, depending on the quenching protocol. We finally mention a simulation of an LJ system[40], which was large enough to include Debye phonons. In this simulation the use of a hard cutoff at $r_c$ (no tapering, "$m = 0$") was compared with $m = 1$ tapering. Interestingly only the in $m = 1$ case non-phononic modes on top of the Debye phonons were observed.

## Resulting scenario for the role of nonphononic excitations in glasses

In a nutshell, the scenario described by the HET and by the GHET, distinguishes (i) between ($i_A$) large, macroscopic, glasses as those investigated experimentally, where the acoustic waves (phonons) extend to zero frequency, and ($i_B$) small systems investigated numerically by molecular dynamics simulations and similar techniques where the phonons exist only above a certain lowest resonance frequency; (ii) between ($ii_A$) "quickly" quenched glasses, such as those obtained by quenching from a high parental temperature in numerical simulations

(here the glass is at criticality $\gamma \approx \gamma_c$), and ($ii_B$) "slowly" quenched glasses, obtained by quenching from a well equilibrated low parental temperature, or in the real world, where the quenching rate is by far slower than in any numerical simulations (here the glass is deeply in the stable region $\gamma_c \gg \gamma$). Within the present description (in terms of HET and GHET) in all these cases a rather different scenario is realized.

In Fig. 1 we summarize, in a cartoon-like fashion, what is expected from our theory.

In Fig. 1a the case of a macroscopically large system far from criticality is considered. This is the situation (almost) *always* found in real experiments. In this figure the reduced DOS $g(\omega)/\omega^2$ is sketched (blue line), for a $3d$ system of infinite size ($N=\infty$), far from criticality ($\gamma \ll \gamma_c$). The high-frequency side is characterized by Anderson-localized modes (also called "locons" following the notation of Allen et al.[65]). The rest of the spectrum is populated by extended modes. This remaining region is divided into (i) a low-frequency regime, where acoustical, only occasionally scattered waves ("phonons"), as considered by Debye, exist, and (ii) an intermediate regime, where the modes are strongly scattered by the disordered structure. In this strongly-scattering regime the intensity of the waves obeys a diffusion equation akin to light in milky glass[66]. This is why Allen et al.[65] call these excitations "diffusons".

It has been suggested[14,22] that the states in this region obey the statistics of random-matrix eigenstates.

In Fig. 1b the reduced density of states, $g(\omega)/\omega^2$ is depicted for $\gamma$ approaching its critical value $\gamma_c$. The blue line represents, as in panel a, the case $\gamma \ll \gamma_c$. The violet, magenta and red lines correspond to increasing disorder, approaching $\gamma \to \gamma_c$. The BP onset shifts to lower and lower frequencies, eventually reaching zero frequency at $\gamma = \gamma_c$. In parallel, the random-matrix type modes, here named 'type-I non-phononic' modes, cover more and more the low-frequency region.

Finally, in Fig. 1c we present the case obtained in numerical simulations of very small systems, where a gap opens in the phonon spectrum. The gap edge is located at the lowest resonance frequency $\omega_o$. This frequency corresponds to the frequency of the transverse phonons in the glass at a wavevector $k = 2\pi/L$, where $L$ is the box size, which gives $\omega_o = 2\pi v_T/L$, where $v_T$ is the transverse sound velocity. As sketched in the Figure, the frequency $\omega_o$ provides the upper limit for the visibility of the type-II spectrum. At higher frequencies these modes hybridize with the waves and probably can no more be distinguished from them.

If the system is quenched from a high parental temperature, i.e. the resulting inherent structure lies high in the potential energy landscape, $\gamma \approx \gamma_c$, the gap in the phonon spectrum exists no more, i.e the non-phononic type-I modes (random matrix eigenstates) extend towards $\omega = 0$. The low-frequency DOS in this case scales as $g(\omega) \sim \omega^2$ like a $d = 3$ Debye spectrum (this case is not sketched in the figure).

If, on the contrary (as reported in of Fig. 1c), the quenching is performed starting from a well equilibrated (supercooled) liquid configuration at low parental temperature, the inherent structure reaches a low value in the potential energy landscape, and the glass is far from criticality. At low frequencies we have both a gap in the phonon spectrum (because of the small system size) *and* a gap in the non-phononic type-I modes (because $\gamma \ll \gamma_c$). Within this gap now emerge the otherwise not visible non-phononic type-II modes.

It is worth to note that the non-phononic type-II modes can be accessed in a very specific situation (small system size) that can be reached only in numerical simulations. As mentioned in the introduction, these modes have been the subject of a large amount of work in the last years, almost all of them reporting a density of states $g(\omega) \sim \omega^s$ with $s = 4$. The GHET predicts that this $s$ value depends on the details of the potential. It turns out that the value of $s$ depends on the tapering function used to smooth the potential around the cutoff in the molecular dynamic simulations, such that the first $m$th derivatives are continuous. For $m = 2$, which is *almost always* used in numerical

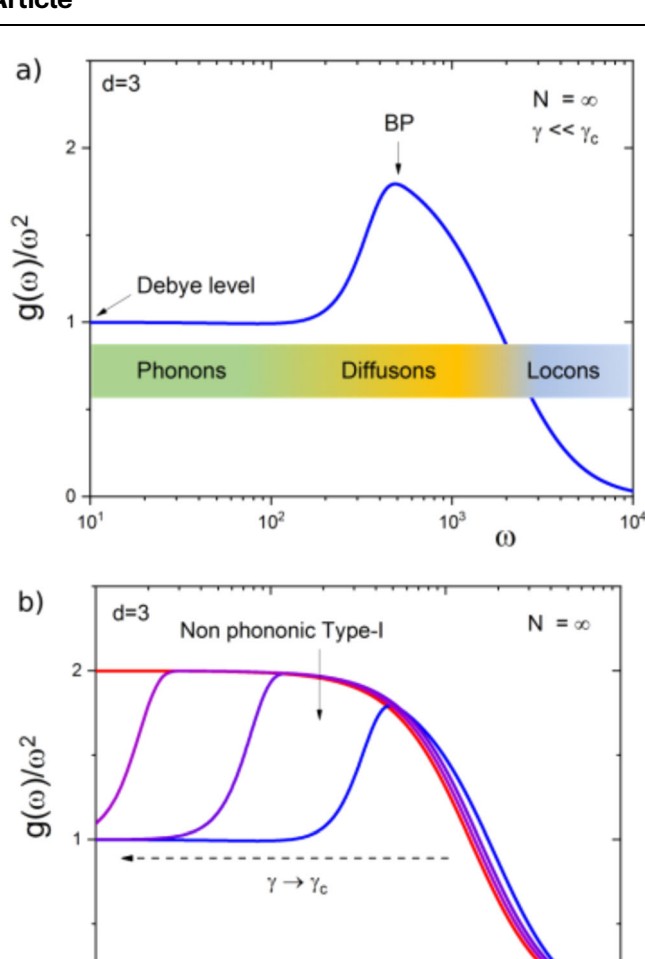

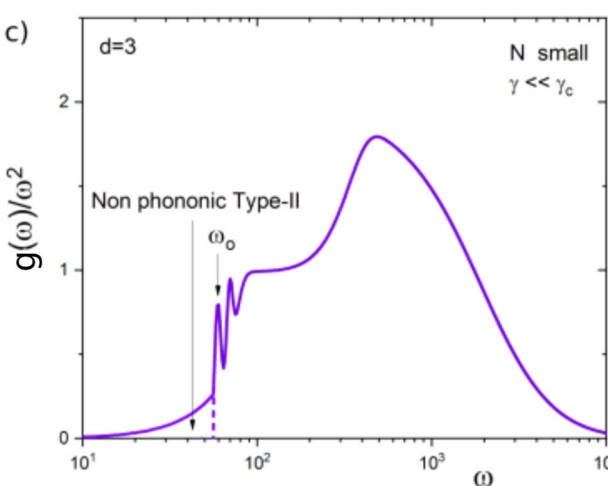

**Fig. 1 | Reduced density of states in different situations. a** Sketch of the reduced DOS, $g(\omega)/\omega^2$ (blue line), for a $3d$ system of infinite size ($N=\infty$), far from criticality ($\gamma \ll \gamma_c$). The high-frequency side is characterized by Anderson-like localized modes (also called `Locons' following the Allen-Feldman[65] notation), while the rest of the spectrum is populated by extended modes. These are divided into a low-frequenncy region where phonon(-like) modes exist (briefly `Phonons') and an intermediate region where the modes are strongly scattered by the disordered structure, and their dynamics is wave diffusion (`Diffusons'). According to the HET, this region starts at the onset of the BP and the modes in this region are of random-matrix type, i.e. similar to eigenstates of random matrices. **b** Sketch of the reduced density of states, $g(\omega)/\omega^2$, for a $d=3$ system of infinite size ($N=\infty$), at different level of criticality. The blue line represents, as in (**a**) the case $\gamma \ll \gamma_c$, while the violet, magenta and red lines correspond to cases with $\gamma \to \gamma_c$. The BP onset shifts to lower and lower frequency, eventually reaching zero frequency at $\gamma = \gamma_c$. In parallel, the random-matrix type modes, here named `type I non-phononic' modes, cover more and more the low-frequency region. **c** Sketch of the reduced density of states, $g(\omega)/\omega^2$ (violet line), for a $d=3$ system, with *finite* size ($N$), and far from criticality ($\gamma \ll \gamma_c$). Due to the finite size of the system, the boundary conditions impose a lowest $k$ value, and consequently a lowest phonon frequency $\omega_o$. Below this value, no phonons are allowed. If—as in the sketch—the system is far from criticality, a gap opens at small frequencies (no phonons *and* no non-phonic type-I modes are allowed in the low frequency region). On the contrary, different numerical simulation studies indicate that in this condition the low-frequency part of the spectrum shows an $\omega^4$ DOS. We have called these excitations "non-phononic type-II modes", and we have shown that their origin can be explained via the generalized HET (see text).

decide not to consider the cases with $m < 2$, where some matrix elements of the Hessian are discontinuous at the truncation point.

Finally, as pointed out above, in the case of a potential with repulsive and attractive parts, which has a minimum at the nearest-neighbour distance—like the Lennard-Jones potential – the GHET predicts that the dominant type-II spectrum scales with $s = 5$, in agreement with recent simulations[52,53].

The GHET, therefore, is in agreement with results previously reported in the literature. The universally observed $s = 4$ is due to a specific choice of tapering made in the numerical simulations, whereas $s = 5$ is generic for potentials with a minimum.

## Numerical simulations

As the GHET gives us precise predictions on the value of $s$ for different tapering shapes, and links this value to the (distribution of the) local stress tensor, we have performed extensive numerical simulations of a model glass to give strength to the theory. We performed the simulations with different tapering functions (specifically, a $C^2$ and a $C^\infty$ function), and measured both the stress distributions and the non-phononic spectra. Both turn out to be in agreement with GHET.

As a microscopic model of glass, we consider a polydisperse mixture of short-range repulsive particles[70]. The diameters $a_i$, with $i = 1, \ldots, N$ the particle's label, are drawn from a power law distribution $P(a)$ with $\langle a \rangle \equiv \int_{a_{min}}^{a_{max}} da\, P(a) a = 1$, with $a_{min} = 0.73$, $a_{max} = 1.62$, and $P(a) = \mathcal{N} a^{-3}$, with $\mathcal{N}$ the proper normalization constant[70-76]. Particles are arranged into a box of side $L$ with periodic boundary conditions. Indicating with $r_1, \ldots, r_N$ a configuration of the system, the mechanical energy is

$$V(\boldsymbol{r}_1, \ldots, \boldsymbol{r}_N) = \frac{1}{2}\sum_{i \neq j} \phi(r_{ij}), \tag{13}$$

with the pair potential $\phi(r_{ij})$

$$\phi(r_{ij}) = \left(\frac{a_{ij}}{r_c}\right)^n \varphi_m(x_{ij})$$
$$\varphi_m(x) = [\varphi(x) - \varphi(1)]\, T_m(x) \tag{14}$$
$$\varphi(x) = x^{-n}.$$

simulations, the prediction is $s = 4$, as observed in the literature. More generally, $s = 3 + 2/m$. As for a correct numerical determination of the Hessian (and thus of the dynamics) $m$ must be $m \geq 2$, $s$ cannot be $>4$. It can be reduced to its minimum value, $s = 3$, if a $C^\infty$-function is used as tapering function.

There exists a large body of simulations with with a potential, which quadratically becomes zero at a radius $r_c$ (see Ref. 67 for references). Such potentials have been used in jamming studies[68,69]. The Hessian of this potential is discontinuous at $r = r_c$. The Hessian, however, on which our analytical theory is based, and from which the DOS is obtained by diagonalization, plays a pivotal role. Therefore we

with $x_{ij} = r_{ij}/r_c$, $r_{ij} = |\mathbf{r}_i - \mathbf{r}_j|$ and $r_c$ is the cutoff distance. In writing (14) we have introduced $a_{ij} \equiv \frac{a_i + a_j}{2}$ and we indicate with $T_m(x)$ the cutoff function (tapering function) which is different from zero only in the interval $x \in [0, 1]$. $T_m(x)$ has the property that it varies as $(1-x)^m$ for $x \to 1$. Consequently the potential goes to zero for $r \to r_c$ as

$$\phi(r_{ij}) \sim (r_{ij} - r_c)^{m+1} \tag{15}$$

This guarantees that the first $m$ derivatives of the potential vanish continuously at the cutoff $r_c$. We specialize our numerical study to the case $m = 2$ and $m = \infty$ adopting the following functions

$$T_2(x) = (1 - x^2)^2, \tag{16}$$

and

$$T_\infty(x) = \exp\left[\frac{x}{2(x-1)}\right]. \tag{17}$$

The cutoff distance $r_c$ is chosen to be larger than the largest particle's diameter $\sigma_{max} = 1.62$ (we set $r_c = 1.955$).

The simulated systems are small enough to allow a direct inspection of the non-phononic modes, be it Type-I or Type-II. Actually, we measure the lowest phonon resonance at $\omega_o \approx 1.5$, and modes are found down to a frequency ten times smaller.

To check the character of these modes, i.e. whether they are extended or localized, we use the fact that in a disordered system eigenvalues cannot be degenerate, which leads to *level repulsion*. In the theory of random matrices[77] this phenomenon is investigated using the statistics of the eigenvalue differences. For delocalized states this statistics is governed by that of the Gaussian Orthogonal Ensemble (GOE), for which the distribution of small eigenvalue differences increases linearly, whereas for localized states one expects a Poissonian (exponentially decaying) distribution.

For evaluating the GOE spectral statistics we use the method reported in[78] where the distribution of the variable $r$ (the ratio of two differences between adjacent eigenvalues) is considered. The variable $r$ is defined as $r = (\lambda_{p+1} - \lambda_p)/(\lambda_{p+2} - \lambda_{p+1})$ $(p = d-1 \ldots [dN-2])$, where $\lambda_p$ are the eigenvalues ordered in ascending order.

Atas et al.[78] have shown that for the GOE case the statistics is well obeyed by their surmise

$$P(r) = \frac{27}{8} \frac{r + r^2}{(1 + r + r^2)^{5/2}}. \tag{18}$$

Figure 2 shows the distribution $P(r)$ for three different frequency regions: lowest 50 eigenvalues (panel A), middle frequency range (B) and highest 50 eigenvalues of a $d = 3$ and $N = 1600$ system. The distributions have been averaged over 900 independent samples. The highest frequency modes, as expected, are localized (the dotted line is an exponential fit to the data), while both middle and low frequency modes show their extended character: they are very well represented by the Atas surmise, Eq. (18), as demonstrated by the parameterless full line, which represents the data very well. It is interesting to note that the level distance statistics is universal, i.e. does not depend on the type of non-phononic excitations: type-I in samples prepared from high $T^*$ (blue symbols), type-II in samples prepared from low $T^*$ (red symbols).

This result confirms that all low-frequency non-phononic modes, type-I and type-II are extended and not localized.

We now turn to the evaluation of the distribution of the local frozen-in stresses. Liquid theory (see Methods M3) predicts the behaviour of the stress distribution $\mathcal{P}(\sigma)$ at low $\sigma$ value. Specifically, Eq. (11) predicts a power law $P(\sigma) \sim \sigma^\alpha$ with exponent $\alpha = \frac{1}{m} - 1$, that is $\alpha = -1/2$ for $m = 2$ and $\alpha = -1$ for $m = \infty$, which are the two simulated cases.

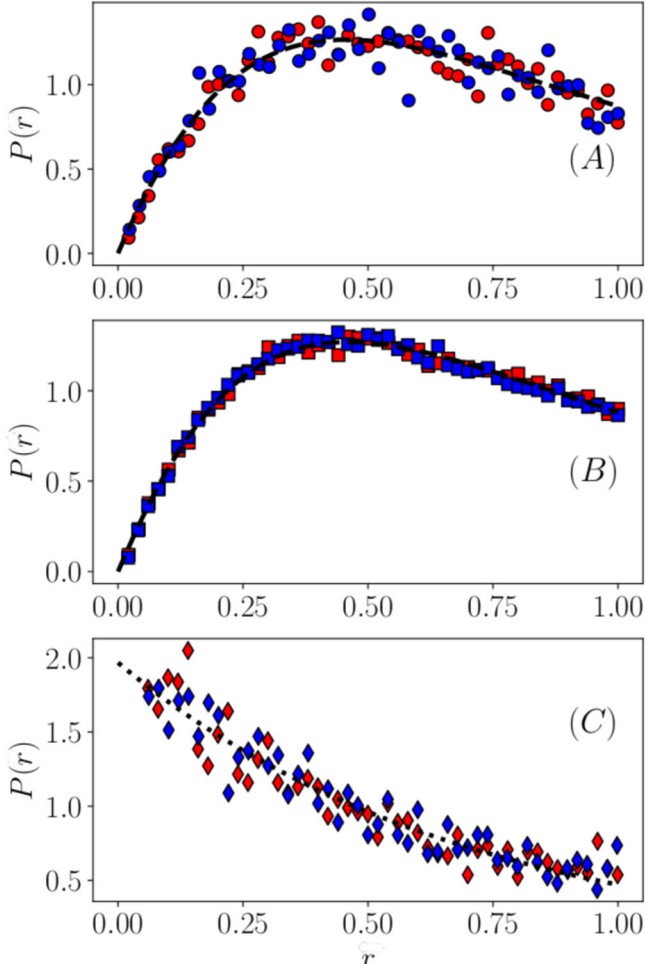

**Fig. 2 | Eigenvalue statistics.** Distribution of the quantity[78] $r = (\lambda_{p+1} - \lambda_p)/(\lambda_{p+2} - \lambda_{p+1})$ $(p = d-1 \ldots dN-2)$. Blue symbols: High parental temperature, red symbols: low parental temperature; Panel **A**: low-frequency modes (lowest 50 modes); panel **B**: middle frequency regime; panel **C**: High-frequency (highest 50) modes. The highest-frequency modes, as expected, are localized (the full line is an exponential fit to the data), while both middle- and low-frequency modes show their extended character, as they are very well represented (full lines) by the Gaussian-Orthogonal-Ensemble surmise, Eq. (18) of Atas et al.[78]. The data have been obtained by evaluating 900 samples.

Figure 3 reports the distributions $\mathcal{P}(\sigma)$ in a double-logarithmic scale for the investigated $m = 2$ (black) and $m = \infty$ (red) cases. The dashed lines represent the expected slopes. The present results clarify how the local stress distributions in the numerical simulation are controlled by the need of introducing a tapering for giving continuity to the interaction potential and its first two derivatives.

As last step we evaluate the integrated density of states (The data quality of this function is much better than that of $g(\omega)$, evaluated by a binning procedure.). from the numerical simulations:

$$\mathcal{F}(\omega) = \int_0^\omega d\omega' \, g(\omega'). \tag{19}$$

It is clear that for a DOS with $g(\omega) \sim \omega^s$ ($s > -1$) the integrated DOS behaves as $\mathcal{F}(\omega) \sim \omega^{s+1}$. The results are reported in Fig. 4. The filled symbols correspond to the "usual" simulations: the tapering function is chosen to ensure the continuity up to the second derivative ($m$=2, Eq. (16)) of the interaction potential, while the open symbols correspond to $m = \infty$ with tapering function (17). In the top panel (A), the quenching is performed starting at a high parental temperature $T^*$, i.e. the

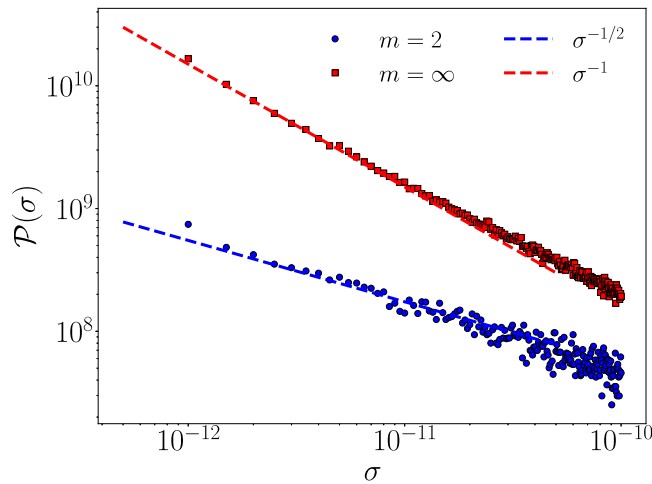

**Fig. 3 | Stress distributions.** Distribution of the (modulus of) the off-diagonal stress components. (A) blue dots, case $m = 2$. The bin size is $10^{-4}$. The dashed line indicates the slope 1/2, as expected according to Eq. (11) for $m = 2$. B red dots, case $m = \infty$. The bin size is $10^{-5}$, smaller than in the previous case as for $m = \infty$ there are much more small stress values. In this case the dashed line indicates the slope 1, again expected according to Eq. (11) for $m = \infty$.

equilibration of the liquid is at $T \gg T_d$. The system is then rapidly quenched, i.e. it does not have enough time to equilibrate the glass structure, which, therefore, is close to marginality. Consequently, as already noticed in literature[42,43] and as predicted by HET, the DOS follows a $\omega^2$, with $s = 2$, scaling, *which is independent of the potential tapering*.

If, on the contrary, the parental temperature $T^*$ is lowered, and approaches $T_d$, the system is far from marginality, a gap opens in the non-phononic type-I spectrum, and the type-II modes now dominate the low-frequency spectrum. This spectrum is predicted by GHET to be affected by the way the potential is tapered. For better transparency we collected the GHET predictions for different values of the tapering index $m$ in Table 1. Indeed, as reported in the bottom panel (B), we observe a different low-frequency behavior for the two different tapering versions: for the $m = 2$ tapering $s = 4$ is reached at low frequencies, and for the $m = \infty$ tapering $s = 3$, as predicted by GHET for the DOS of the type-II modes.

In other words: The main result of Fig. 4 is the *insensitivity* of the DOS on the tapering exponent $m$ for high parental temperature $T^*$, as opposed to a *sensitivity* on $m$ for low $T^*$. The observed low-frequency exponents compare well with the predictions of HET and GHET: $s = 2$ for high $T^*$, signifying type-I excitations at marginality, and $s = 4, 3$ for low $T^*$, corresponding to type-II excitations, which depend on the stress distributions for tapered potentials with $m = 2, \infty$.

## Discussion

We have used the heterogeneous elasticity theory (HET) and its generalization (GHET) in order to clarify the behaviour of harmonic vibrational excitations in disordered systems.

In the present study we clarify the interplay between system stability (controlled by the disorder parameter $\gamma$, i.e. the normalized variance of the elastic constant heterogeneity) and the system size in determining the DOS shape at low frequency.

As predicted by HET, in large macroscopic systems (as real glasses are), the disorder parameter controls the frequency position $\omega_{BP}$ of the boson peak (BP), and $\omega_{BP}$ marks the frequency border between the phonon realm at low frequency and that of the non-phononic type-I excitations at $\omega > \omega_{BP}$. If the disorder is increased—i.e. the stability is decreased—the system approaches the marginal state. Here the BP

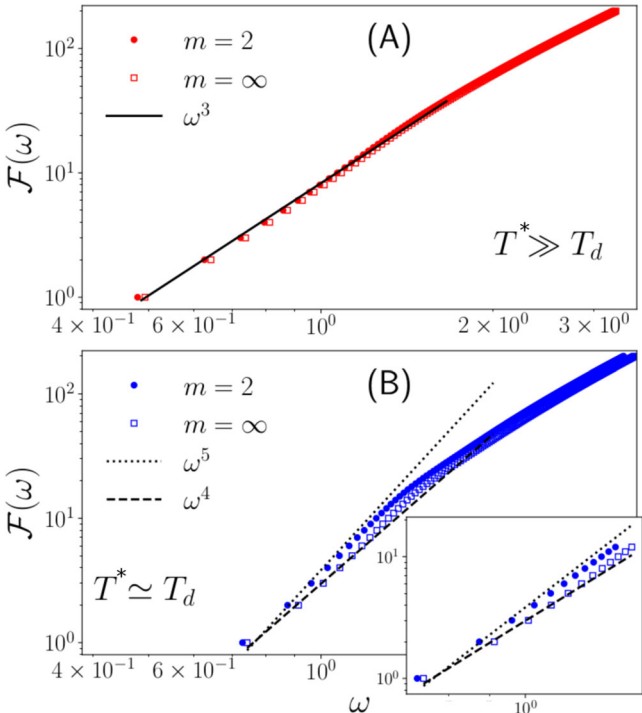

**Fig. 4 | Dependence of the spectra for different tapering versions.** Numerical results for the integrated DOS $\mathcal{F}(\omega) = \int_0^\omega d\omega' g(\omega')$ of a 'small' ($N = 1000$) system obtained for a high ($T \gg T_D$, panel A) and low ($T \simeq T_d$, B) parental temperature, where $T_d$ is the temperature of dynamical arrest. The full symbols report the 'usual' tapering case ($m = 2$), while the open ones correspond to $m = \infty$, Eq. (17), introduced in this work, in order to test the GHET. For high parental temperature (**A**) the low-frequency slope of the DOS $s = 2$ for both tapering versions, corresponding to the marginal type-I spectrum, while at low parental temperature (**B**) where a gap opens in the type-I-mode spectrum, the DOS is due to the type-II excitations, which exhibit an $m$ dependent low-frequency scaling: $s = 4$ for $m = 2$ and $s = 3$ for $m = \infty$.

**Table 1 | Spectral exponents for the different tapering schemes**

| Quantity | Scaling | $m = 1$ | $m=2$ | $m = \infty$ |
|---|---|---|---|---|
| $\mathcal{P}(\sigma)$ | $\sim \sigma^{-1+\frac{1}{m}}$ | $\sigma^0$ | $\sigma^{-\frac{1}{2}}$ | $\sigma^{-1}$ |
| $\rho_{\varepsilon\eta}(\lambda)$ | $\sim \lambda^{1+\frac{1}{m}}$ | $\lambda^2$ | $\lambda^{\frac{3}{2}}$ | $\lambda^1$ |
| $g_{\varepsilon\eta}(\omega)$ | $\sim \omega^{3+\frac{2}{m}}$ | $\omega^5$ | $\omega^4$ | $\omega^3$ |
| $\mathcal{F}_{\varepsilon\eta}(\omega)$ | $\sim \omega^{4+\frac{2}{m}}$ | $\omega^6$ | $\omega^5$ | $\omega^4$ |

We show the results for $\mathcal{P}(\sigma)$, the level density $\rho(\lambda)$, the DOS $g(\omega)$ and the integrated DOS $\mathcal{F}(\omega) = \int_0^\omega d\omega' \, g(\omega')$ for the case of a potential with cutoff and tapering function of order $m$.

moves to zero frequency, the phonons disappear, and the type-I modes dominate the whole frequency region.

When the system size becomes small enough, a new frequency scale enters into the game: the lowest phonon resonance, $\omega_o$, which fixes the lower frequency boundary of waves allowed by the system ($\omega > \omega_o$). If the system is marginally stable, or close to marginal stability, below $\omega_o$ the type-I excitations dominate the scene. Their DOS is – like that of the phonons in $d = 3$ – proportional to $\omega^2$ in any dimension $d$, although with a different prefactor. If, on the contrary, the glass is highly stable ($\gamma \ll \gamma_c$) a gap opens in the type-I non-phononic excitations spectrum: the type-I excitations only exists for $\omega > \omega_c$. It is in these conditions (stable small glass) that the type II non-phononic excitations can show up below both $\omega_c$ and $\omega_o$.

In other words—if the overwhelming phonon (Propagon) modes, whose DOS is $g(\omega) \simeq \omega^{d-1}$, are removed from the low-frequency realm

by numerically investigating small systems – in addition to the type-I modes situated above the BP frequency – a new kind of excitations emerges from the background. It is worth to note that type-II non-phononic excitations are always present, but they are hidden by the presence of waves or by the presence of the type-I modes. When the waves are "removed" (small systems) and the type-I modes are removed as well (well equilibrated systems), the type-II non-phononic modes become visible.

At this point we note that that the local stresses, which give rise to the type-II modes involve a length scale of a few interatomic distance, whereas the heterogeneous elasticity, responsible for the type-I modes involves the mesoscopic length scale of the coarse-graining procedure. To investigate the difference of the two types of non-phononic excitations in more detail is an interesting task for future research. Similarly, it will be very interesting to investigate, how the local stresses and the associated type-II modes enter into the vibrational spectrum of macroscopic samples, and, in particular, how they may influence the boson peak.

The depicted scenario for small samples allows to clarify recent numerical findings, where a $\omega^s$ DOS has been observed, with $s$ depending on the specific parental temperature: $s = 4$ for minima deep in the potential energy landscape (type-II modes in well equilibrated small system), and $s \to 2$ on increasing the potential energy of the minimum (type-I modes in badly equilibrated small system).

After having clarified the scenario of the excitations existing in glasses, we have investigated the nature of the type-II non-phononic ones. By performing a proper continuum limit for the harmonic dynamics of the glass, we found that in addition to the strain field considered in heterogeneous-elasticity theory (HET), one has to take the non-irrotational part of the atomic displacement pattern into account. The dynamics of these local vorticities are governed by the statistics of the local frozen-in stresses. The local vorticities act as local oscillators, hybridized with the waves and the type-I modes.

The generalization of HET (GHET) predicts that the scaling of the DOS of type-II non-phononic modes at low frequency is related to the distribution of the local stresses. The low-frequency scaling of the type-II excitations is predicted to be non-universal and related via the stress distributions to the statistics of the interatomic forces, giving a one-to-one relation between the potential statistics and the spectrum. These predictions are verified in our soft-sphere simulation. Both GHET and the simulation show that in simulations type-II modes are governed by the type of the smooth cutoff (tapering), employed in the simulations, i.e. they are *not* an intrinsic glass property.

For potentials with a minimum near the first-nearest-neighbor potential GHET predicts a scaling with $s = 5$, which is instead a generic property of the system.

At the end we would like to emphasize that the present theory, along with the previous numerical results, obtained by diagonalizing the Hessian matrix (e.g.[38,38,39,41–44,67]), is inherently harmonic. Anharmonic effects, which certainly contribute to experimentally measured spectra (e.g.[4–9]) and spectra obtained by evaluating correlation functions, using molecular-dynamics simulations (e.g.[25–28]), are not taken into account. The harmonic spectra pertain to zero temperature, while the anharmonic effects vanish in this limit. However, the experimentally measured non-phononic spectra in the THz range (boson peak) are reportedly dominated by the harmonic interaction[10,65].

## Methods

### Heterogeneous elasticity theory
Here we review heterogeneous-elasticity theory (HET)[19,28,34], which accounts for the type-I nonphononic excitations.

Heterogeneous elasticity theory is just ordinary elasticity theory with a spatially fluctuating shear modulus G($\mathbf{R}$). The equations of

motion for the displacement fields $\mathbf{u}(\mathbf{R},t)$ are

$$\rho_m \ddot{u}^\alpha(\mathbf{R},t) = \sum_\beta \frac{\partial}{\partial \beta} \sum_{\gamma\delta} C^{\alpha\gamma\delta\beta}(\mathbf{R})\varepsilon^{\gamma\delta}(\mathbf{R},t) \tag{20}$$

Within HET one discards the non-isotropic terms and assumes that the bulk modulus $K$ does not exhibit spatial fluctuations[28,34]. This gives

$$\rho_m \ddot{u}^\alpha(\mathbf{R},t) = K\frac{\partial}{\partial x_\alpha}\mathrm{Tr}\{\varepsilon(\mathbf{R},t)\} + \sum_\beta \frac{\partial}{\partial x_\beta}2G(\mathbf{R})\widehat{\varepsilon}^{\alpha\beta}(\mathbf{R},t) \tag{21}$$

where we have defined the traceless strain tensor

$$\widehat{\varepsilon}^{\alpha\beta}(\mathbf{R},t) \doteq \varepsilon^{\alpha\beta}(\mathbf{R},t) - \frac{1}{3}\mathrm{Tr}\{\varepsilon(\mathbf{R},t)\} \tag{22}$$

Fourier-transformed into the frequency regime, Eq. (21) takes the form

$$\begin{aligned}&-\omega^2\rho_m u^\alpha(\mathbf{R},\omega)\\&= K\frac{\partial}{\partial x_\alpha}\mathrm{Tr}\{\varepsilon(\mathbf{R},\omega)\} + \sum_\beta\frac{\partial}{\partial x_\beta}2G(\mathbf{R})\widehat{\varepsilon}^{\alpha\beta}(\mathbf{R},\omega)\end{aligned} \tag{23}$$

The stochastic Helmholtz equation (23) can be solved with the help of a mean-field theory, the self-consistent Born approximation[19,20,28,34] (SCBA). The SCBA arises as a saddle point within a replica-field theoretic treatment. The solution amounts to replacing the true medium with spatially fluctuating shear modulus $G(\mathbf{r}) = G_0 - \Delta G(\mathbf{r})$ by a complex, frequency-dependent one $G(z)$ with $z = \omega^2 + i0$. The input is the mean value $G_0 = \langle G(\mathbf{r})\rangle$ and the variance $\langle(\Delta G)^2\rangle$, and the output is $G(z) = G_0 - \Sigma(z)$, where $\Sigma(z)$ is the self energy.

The SCBA equations for $\Sigma(z)$ are formulated in terms of the longitudinal ($L$) and transverse ($T$) Green's functions, which are given by

$$\mathcal{G}_{(L,T)}(k,z) = \frac{1}{-z + k^2 v_{(L,T)}^2(z)} \tag{24}$$

As usual one has to add an infinitesimal positive imaginary part to the spectral variable $\lambda = \omega^2$: $z \doteq \lambda + i0$. The quantities $v_L(z)$, $v_T(z)$ are effective, complex, frequency-dependent sound velocities, given by

$$v_L^2(z) = \frac{1}{\rho_m}\left[K + \frac{4}{3}(G_0 - \Sigma(z))\right] \tag{25}$$

$$v_T^2(z) = \frac{1}{\rho_m}[G_0 - \Sigma(z)] \tag{26}$$

Here $K$ the bulk modulus that we assume to be uniform along the sample. This expression holds in $d = 3$, in other spatial dimensions the factor 4/3 in the RHS of Eq. (25) will be different. The quantity $M = K + 4/3G_o$ is the mean longitudinal modulus, thus we can define $M(z) = K + 4/3(G_o - \Sigma(z))$ as the generalised longitudinal modulus. The self consistent HET-SCBA equation for the self energy $\Sigma(z)$ turns out to be[79]:

$$\Sigma(z) = \int \frac{d^3\mathbf{k}}{(2\pi)^3}C(k)k^2\left(\frac{2}{3}\mathcal{G}_L(k,z) + \mathcal{G}_T(k,z)\right) \tag{27}$$

Here $C(k)$ is the Fourier transform of the correlation function function $C(r)$

$$C(\mathbf{r}) = \langle\Delta G(\mathbf{r}_0 + \mathbf{r})\Delta G(\mathbf{r}_0)\rangle = \langle(\Delta G)^2\rangle f(r) \tag{28}$$

of the fluctuations $\Delta G(\boldsymbol{r}) = G(\boldsymbol{r}) - \langle G \rangle$ of the elastic shear moduli. We assume that these correlations are short-ranged, i.e. $f(k=0)$ is supposed to be finite. Because the inverse correlation length $\xi^{-1}$ acts effectively as an ultraviolett cutoff, we schematically use

$$f(k) = f_0 \theta(k_\xi - k), \tag{29}$$

where $\theta(x)$ is the Heaviside step function and $k_\xi$ is inversely proportional to the correlation length. From the condition

$$1 = f(r=0) = \int \frac{d^3 \boldsymbol{k}}{(2\pi)^3} f(k) \tag{30}$$

we obtain

$$f_0 = \frac{6\pi^2}{k_\xi^3} \tag{31}$$

We now introduce dimensionless variables: $q = k/k_\xi$, $\tilde{z} = z^2 \rho_m / G_0 k_\xi^2$, $\widetilde{\Sigma} = \Sigma/G_0$, $\tilde{K} = K/G_0$, $\tilde{M}(\tilde{z}) = \tilde{K} + \frac{4}{3}[1 - \widetilde{\Sigma}(\tilde{z})]$, $\tilde{\mathcal{G}}_{(L,T)}(q, \tilde{z}) = \mathcal{G}_{(L,T)}(k, z)$. We also define the dimensionless disorder parameter:

$$\gamma \doteq \frac{1}{G_0^2} \langle (\Delta G)^2 \rangle. \tag{32}$$

In terms of these quantities the self-consistent equation (27) takes the form

$$\begin{aligned}\widetilde{\Sigma}(\tilde{z}) &= 3\gamma \int_0^1 dq\, q^4 \left( \frac{2}{3} \tilde{\mathcal{G}}_L(q, \tilde{z}) + \tilde{\mathcal{G}}_T(q, \tilde{z}) \right) \\ &= \gamma \left( \frac{2}{3} \frac{1}{\tilde{M}(\tilde{z})} \left[1 + \tilde{z}\tilde{\mathcal{G}}_L(\tilde{z})\right] + \frac{1}{1 - \widetilde{\Sigma}(\tilde{z})} \left[1 + \tilde{z}\tilde{\mathcal{G}}_T(\tilde{z})\right] \right)\end{aligned} \tag{33}$$

with

$$\tilde{\mathcal{G}}_{(L,T)}(\tilde{z}) \doteq 3 \int_0^1 dq\, q^2\, \tilde{\mathcal{G}}_{(L,T)}(q, \tilde{z}) \tag{34}$$

The DOS $g_\varepsilon(\omega)$ and the level density $\rho_\varepsilon(\tilde{z})$ are given by

$$g_\varepsilon(\omega) = 2\omega \rho_\varepsilon(\tilde{z}) = \frac{2\omega}{3\pi} \text{Im}\{\tilde{\mathcal{G}}_L(z) + 2\tilde{\mathcal{G}}_T(z)\}. \tag{35}$$

We introduced the subscript $\varepsilon$ in order to distinguish the HET-type-I spectra from those due to the type-II excitations, which we call $g_{\varepsilon\eta}(\omega)$ and $\rho_{\varepsilon\eta}(\lambda)$.

We now multiply by the factor $1 - \widetilde{\Sigma}(\tilde{z})$ and define a function $\tilde{B}(\tilde{z}) \doteq \frac{2}{3}[1 - \widetilde{\Sigma}(\tilde{z})]/\tilde{M}(\tilde{z})$ and obtain

$$\widetilde{\Sigma}(\tilde{z})[1 - \widetilde{\Sigma}(\tilde{z})] = \left( \left[1 + \tilde{B}(\tilde{z})\right] + \tilde{z}\tilde{\mathcal{G}}_0(\tilde{z}) \right) \tag{36}$$

with

$$\tilde{\mathcal{G}}_0(\tilde{z}) \doteq \gamma \left( \tilde{\mathcal{G}}_T(\tilde{z}) + B(\tilde{z})\tilde{\mathcal{G}}_L(\tilde{z}) \right) \tag{37}$$

Because $\tilde{B}(\tilde{z})$ depends only weakly on $\tilde{z}$, for small $\tilde{z}$ we may set $\tilde{z}$ in this function equal to zero. This gives

$$\widetilde{\Sigma}(\tilde{z})[1 - \widetilde{\Sigma}(\tilde{z})] = \left( \underbrace{\gamma\left[1 + \tilde{B}(0)\right]}_{\tilde{\gamma}} + \tilde{z}\tilde{\mathcal{G}}_0(\tilde{z}) \right) \tag{38}$$

we may now solve this quadratic equation to obtain

$$\widetilde{\Sigma}(\tilde{z}) = \widetilde{\Sigma}_c(0) + \sqrt{\tilde{\gamma}_c - \tilde{\gamma} - \tilde{z}\tilde{\mathcal{G}}_0(\tilde{z})} \tag{39}$$

with $\tilde{\gamma}_c = \frac{1}{4}$ and $\widetilde{\Sigma}_c(0) = \widetilde{\Sigma}(0)|_{\tilde{\gamma}=\tilde{\gamma}_c} = \frac{1}{2}$. For $\tilde{\gamma} > \tilde{\gamma}_c$ obviously $\Sigma(0)$ becomes imaginary, which introduces a finite DOS for $\tilde{z} \leq 0$, i.e. an instability. This instability is due to the presence of too many negative local shear moduli, which in SCBA have been assumed to obey Gaussian statistics.

In very small samples the integrals in Eqs. (27) and (34) have to be replaced by discrete sums over wavevectors $\boldsymbol{k} = (k_x, k_y, k_z)$ with $k_\alpha = 2\pi n_\alpha/L$, $n_\alpha \in \mathbb{Z}$. For small frequencies the resonance conditions $\tilde{\omega} = k v_{(L,T)}(0)$ in the denominator of the Green's functions cannot be met (the sample is too small for carrying long-wavelength waves), and the Green's functions $\mathcal{G}_{(L,T)}$, and hence $\mathcal{G}_0$ become real quantities. This, in turn leads to a *gap* in the DOS for $\tilde{\gamma} < \tilde{\gamma}_c$:

$$\rho_\varepsilon(\tilde{\lambda}) \begin{cases} = 0 & \text{for } \tilde{\lambda} < \tilde{\lambda}_c \\ \sim \sqrt{\tilde{\lambda} - \tilde{\lambda}_c} & \text{for } \tilde{\lambda} \geq \tilde{\lambda}_c \end{cases} \tag{40}$$

with $\tilde{\lambda}_c = [\tilde{\gamma}_c - \tilde{\gamma}]/\mathcal{G}_0(0)$. At marginal stability $\tilde{\lambda}_c = 0$ this leads to $\rho(\tilde{\lambda}) \sim \tilde{\lambda}^{1/2}$ and consequently to a DOS $g(\omega) \sim \omega^2$.

In macroscopically large, stable samples the gap is filled by the Debye phonons. The onset of the imaginary part of $\widetilde{\Sigma}(z)$ at $\tilde{\lambda}_c$ corresponds to the boson peak[19,20,28,34], and the states above $\tilde{\lambda}_c$ are irregular random-matrix states.

## Derivation of Generalized Heterogeneous-Elasticity Theory (GHET)

We confine our treatment to three dimensions; the generalization to two dimensions is straightforward.

We start with the harmonic expansion of the total energy

$$\begin{aligned} E &= T + V(\boldsymbol{r}_1, \ldots, \boldsymbol{r}_N) \\ &= T + V_o - \sum_i \boldsymbol{f}_i \cdot \boldsymbol{u}_i + \frac{1}{2} \sum_{ij} \boldsymbol{u}_i \cdot \overset{\leftrightarrow}{\boldsymbol{H}}_{ij} \cdot \boldsymbol{u}_j \end{aligned} \tag{41}$$

Here $\boldsymbol{u}_i$ are infinitesimal displacements from the equilibrium position $\boldsymbol{r}_i$, the $\boldsymbol{f}_i$ are local forces, defined by

$$f_i^\alpha = -\frac{\partial}{\partial r_i^\alpha} V(\boldsymbol{r}_1, \ldots, \boldsymbol{r}_N), \tag{42}$$

and $\overset{\leftrightarrow}{\boldsymbol{H}}_{ij}$ is the Hessian matrix

$$H_{ij}^{\alpha\beta} = \frac{\partial}{\partial r_i^\alpha} \frac{\partial}{\partial r_j^\beta} V(\boldsymbol{r}_1, \ldots, \boldsymbol{r}_N) \tag{43}$$

At equilibrium the local forces $\boldsymbol{f}_i$ are zero, so we discard them from the discussion. We now assume that $V(\boldsymbol{r}_1, \ldots, \boldsymbol{r}_N)$ can be represented as a sum over pairs with a pair potential $\phi(r)$

$$V(\boldsymbol{r}_1, \ldots, \boldsymbol{r}_N) = \sum_{(i,j)} \phi(r_{ij}) \tag{44}$$

where $\sum_{(i,j)} = \frac{1}{2}\sum_{i \neq j}$ denotes a sum over pairs $(i,j)$, and $r_{ij} = |\boldsymbol{r}_{ij}| = |\boldsymbol{r}_i - \boldsymbol{r}_j|$. We further assume that $\phi(r)$ has a finite range $r_c$.

In terms of the pair potential the Hessian $\vec{H}_{ij}$ can be represented by

$$H_{ij}^{\alpha\beta} = -K_{ij}^{\alpha\beta}\left(1 - \delta_{ij}\right) + \left(\sum_{\ell \neq i} K_{i\ell}^{\alpha\beta}\right)\delta_{ij} \tag{45}$$

with the force constants

$$\begin{aligned}
K_{ij}^{\alpha\beta} &= \left[\phi''(r_{ij}) - \frac{1}{r_{ij}}\phi'(r_{ij})\right]\frac{r_{ij}^\alpha r_{ij}^\beta}{r_{ij}^2} + \frac{1}{r_{ij}}\phi'(r_{ij})\delta_{\alpha\beta} \\
&\doteq \phi^{(1)}(r_{ij})\, r_{ij}^\alpha r_{ij}^\beta + \phi^{(2)}(r_{ij})\,\delta_{\alpha\beta}
\end{aligned} \tag{46}$$

with implicit definition of the functions $\phi^{(1)}(r_{ij})$ and $\phi^{(2)}(r_{ij})$.

With these definitions the harmonic part of the potential energy can be represented by

$$V_H(\boldsymbol{r}_1, \ldots, \boldsymbol{r}_N) = \sum_{(i,j)} V_{ij} \tag{47}$$

with

$$\begin{aligned}
V_{ij} &= \frac{1}{2}\sum_{\alpha\beta} u_{ij}^\alpha K_{ij}^{\alpha\beta} u_{ij}^\beta \\
&= \frac{1}{2}\left(\phi^{(1)}(r_{ij})\sum_{\alpha\beta} r_{ij}^\alpha r_{ij}^\beta u_{ij}^\alpha u_{ij}^\beta + \phi^{(2)}(r_{ij})\, u_{ij}^2\right) \\
&\doteq V_{ij}^{(1)} + V_{ij}^{(2)}.
\end{aligned} \tag{48}$$

Here $u_{ij}^\alpha = u_i^\alpha - u_j^\alpha$.

We now go over to a continuum description by introducing difference and center-of-mass variables $\boldsymbol{r}_{ij} = \boldsymbol{r}_i - \boldsymbol{r}_j$ and $\boldsymbol{R}_{ij} = \frac{1}{2}\left[\boldsymbol{r}_i + \boldsymbol{r}_j\right]$ and interpreting $\boldsymbol{R}_{ij} \doteq \boldsymbol{R}$ as the local vector of the continuum theory. We make a Taylor expansion of $\boldsymbol{u}(\boldsymbol{r}_i)$ around $\boldsymbol{R}$

$$\begin{aligned}
\boldsymbol{u}(\boldsymbol{r}_i) &= \boldsymbol{u}(\boldsymbol{R}) + \left([\boldsymbol{r}_i - \boldsymbol{R}] \cdot \nabla\right)\boldsymbol{u}(\boldsymbol{R}) \\
&= \boldsymbol{u}(\boldsymbol{R}) + \frac{1}{2}\left(\boldsymbol{r}_{ij} \cdot \nabla\right)\boldsymbol{u}(\boldsymbol{R})
\end{aligned} \tag{49}$$

It follows

$$\frac{1}{2}\left[\boldsymbol{u}(\boldsymbol{r}_i) + \boldsymbol{u}(\boldsymbol{r}_j)\right] = \boldsymbol{u}(\boldsymbol{R}) \tag{50}$$

and

$$u^\alpha(\boldsymbol{r}_i) - u^\alpha(\boldsymbol{r}_j) = \sum_\gamma r_{ij}^\gamma u_{\alpha|\gamma}(\boldsymbol{R}) = u_{ij}^\alpha \tag{51}$$

with abbreviation $u_{\alpha|\gamma} \doteq \partial_\gamma u^\alpha$.

We now introduce symmetrized and antisymmetrized spatial derivatives

$$\begin{aligned}
\varepsilon^{\alpha\gamma}(\boldsymbol{R}) &= \frac{1}{2}\left[u_{\gamma|\alpha}(\boldsymbol{R}) + u_{\alpha|\gamma}(\boldsymbol{R})\right] \\
\eta^{\alpha\gamma}(\boldsymbol{R}) &= \frac{1}{2}\left[u_{\gamma|\alpha}(\boldsymbol{R}) - u_{\alpha|\gamma}(\boldsymbol{R})\right]
\end{aligned} \tag{52}$$

or

$$\begin{aligned}
u_{\alpha|\gamma}(\boldsymbol{R}) &= \varepsilon^{\alpha\gamma}(\boldsymbol{R}) - \eta^{\alpha\gamma}(\boldsymbol{R}) \\
u_{\gamma|\alpha}(\boldsymbol{R}) &= \varepsilon^{\alpha\gamma}(\boldsymbol{R}) + \eta^{\alpha\gamma}(\boldsymbol{R})
\end{aligned} \tag{53}$$

Here $\varepsilon^{\alpha\gamma}(\boldsymbol{R})$ are local strains and $\eta^{\alpha\gamma}(\boldsymbol{R})$ are local vorticities. Because this tensor has only three independent entries, one can define a *vorticity vector*

$$\boldsymbol{\eta}(\boldsymbol{R}) = \frac{1}{2}[\nabla \times \boldsymbol{u}(\boldsymbol{R})] = \begin{pmatrix} \eta^{yz}(\boldsymbol{R}) \\ \eta^{zx}(\boldsymbol{R}) \\ \eta^{xy}(\boldsymbol{R}) \end{pmatrix} \tag{54}$$

We define now the symmetric part of the local (Born-type) elastic constants as

$$B^{\alpha\beta\gamma\delta}(\boldsymbol{R}) \doteq \frac{1}{\Omega_Z}\phi^{(1)}(r_{ij}) r_{ij}^\alpha r_{ij}^\beta r_{ij}^\gamma r_{ij}^\delta\Big|_{\boldsymbol{R}=\boldsymbol{R}_{ij}} \tag{55}$$

and the stress tensor as

$$\sigma^{\gamma\delta}(\boldsymbol{R}) \doteq \frac{1}{\Omega_Z}\phi^{(2)}(r_{ij}) r_{ij}^\gamma r_{ij}^\delta\Big|_{\boldsymbol{R}=\boldsymbol{R}_{ij}} \tag{56}$$

Here $1/\Omega_Z N(Z-1)/2\Omega$ is the average number of pairs within the interaction range per volume, where $\Omega$ is the total volume and $Z$ is the average number of neighbors within the interaction range.

With these definitions we can define a local potential energy density as

$$\mathcal{V}(\boldsymbol{R}) = \mathcal{V}^{(1)}(\boldsymbol{R}) + \mathcal{V}^{(2)}(\boldsymbol{R}) \tag{57}$$

with

$$\begin{aligned}
\mathcal{V}^{(1)}(\boldsymbol{R}) &\doteq \frac{1}{\Omega_Z} V_{ij}^{(1)}\Big|_{\boldsymbol{R}=\boldsymbol{R}_{ij}} \\
&= \frac{1}{2}\sum_{\alpha\beta\gamma\delta} B^{\alpha\beta\gamma\delta}(\boldsymbol{R})\varepsilon^{\alpha\gamma}(\boldsymbol{R})\varepsilon^{\beta\delta}(\boldsymbol{R})
\end{aligned} \tag{58}$$

$$\begin{aligned}
\mathcal{V}^{(2)}(\boldsymbol{R}) &\doteq \frac{1}{\Omega_Z} V_{ij}^{(2)}(\boldsymbol{R})\Big|_{\boldsymbol{R}=\boldsymbol{R}_{ij}} \\
&= \frac{1}{2}\sum_{\alpha\gamma\delta} \sigma^{\gamma\delta}(\boldsymbol{R}) u_{\alpha|\gamma}(\boldsymbol{R}) u_{\alpha|\delta}(\boldsymbol{R})
\end{aligned} \tag{59}$$

As noted by Alexander[57,63], the term $\mathcal{V}^{(2)}$ violates local rotation invariance. Therefore in the literature[55,62] a symmetrization procedure has been applied to this term and the symmetrized non-irrotational term is then incorporated into the symmetric potential energy density, that we define $\widetilde{\mathcal{V}}_{ij}^{(1)}$, in the following way:

$$\widetilde{\mathcal{V}}^{(1)}(\boldsymbol{R}) \doteq \frac{1}{2}\sum_{\alpha\beta\gamma\delta} C_{ij}^{\alpha\beta\gamma\delta}(\boldsymbol{R})\varepsilon^{\alpha\gamma}(\boldsymbol{R})\varepsilon^{\beta\delta}(\boldsymbol{R}) \tag{60}$$

with

$$\begin{aligned}
C_{ij}^{\alpha\beta\gamma\delta} &\doteq B_{ij}^{\alpha\beta\gamma\delta} + \frac{1}{4}\left(\sigma^{\gamma\delta}\delta_{\alpha\beta} + \sigma^{\alpha\delta}\delta_{\gamma\beta} + \sigma^{\gamma\beta}\delta_{\alpha\delta} + \sigma^{\alpha\beta}\delta_{\gamma\delta}\right) \\
&\doteq B_{ij}^{\alpha\beta\gamma\delta} + \text{Sy}\{\sigma^{\gamma\delta}\delta_{\alpha\beta}\}
\end{aligned} \tag{61}$$

Here we omitted the arguments ($\boldsymbol{R}$) for brevity, which we do from now on. The remaining part of the irrotational term is then

$$\begin{aligned}
\widetilde{\mathcal{V}}^{(2)} &\doteq \frac{1}{2}\sum_{\alpha\gamma\delta} \sigma^{\gamma\delta}\eta^{\alpha\gamma}\left(\eta^{\alpha\delta} - \varepsilon^{\alpha\delta}\right) \\
&\doteq \widetilde{\mathcal{V}}_{\eta\eta}^{(2)} + \widetilde{\mathcal{V}}_{\eta\varepsilon}^{(2)}
\end{aligned} \tag{62}$$

In terms of the vorticity vector defined in Eq. (54) the two non-irrotational parts of the potential-energy density take the form

$$\widetilde{\mathcal{V}}^{(2)}_{[\eta\eta]ij} = \frac{1}{2}\left(\mathrm{Tr}\{\sigma_{ij}\}\eta^2 - \sum_{\gamma\mu}\sigma^{\gamma\delta}_{ij}\eta^\gamma\eta^\delta\right) \tag{63}$$

$$\widetilde{\mathcal{V}}^{(2)}_{[\eta\varepsilon]ij} = \frac{1}{2}\boldsymbol{\tau}\cdot\boldsymbol{\eta} \tag{64}$$

with the coupling vector

$$\boldsymbol{\tau} = \begin{pmatrix} \sum_\gamma(\sigma^{yy}\varepsilon^{yz} - \varepsilon^{yy}\sigma^{yz}) \\ \sum_\gamma(\sigma^{zy}\varepsilon^{yx} - \varepsilon^{zy}\sigma^{yx}) \\ \sum_\gamma(\sigma^{xy}\varepsilon^{yy} - \varepsilon^{xy}\sigma^{yy}) \end{pmatrix} \tag{65}$$

It is now obvious that the vorticity field $\boldsymbol{\eta}(\boldsymbol{R})$ is exclusively associated with the local stresses $\overleftrightarrow{\sigma}$. Because these stresses are spatially bounded defects in the glass[57,80], we now give them the "defect label" $\ell$, and call the associated vorticity fields $\boldsymbol{\eta}_\ell(\boldsymbol{R})$. We also treat local stresses as traceless, and their sign to be independent of the sign of the potential derivative $\phi'(\boldsymbol{r})$. The latter property is due to the external pressure exerted to the system by the boundary condition.

The Lagrangian density in terms of the two vector variables $\boldsymbol{u}(\boldsymbol{R},t)$ and $\boldsymbol{\eta}_\ell(\boldsymbol{R},t)$ is given by

$$\mathcal{L}_{\mathrm{GHET}} = \mathcal{L}_{\mathrm{HET}} + \Delta\mathcal{L}_{\mathrm{GHET}} \tag{66}$$

with

$$\mathcal{L}_{\mathrm{HET}} = \frac{\rho}{2}[\dot{u}(\boldsymbol{R})]^2 - \mathcal{V}^{(1)} \tag{67}$$

where $\rho$ is the mass density, and

$$\Delta\mathcal{L}_{\mathrm{GHET}} = \sum_\ell\left(\mathcal{T}_{[\eta]\ell} - \mathcal{V}_{[\eta\eta]\ell} - \mathcal{V}_{[\varepsilon\eta]\ell}\right) \tag{68}$$

with the kinetic-energy density of the vorticities

$$\mathcal{T}_{[\eta]\ell} = \frac{1}{2}\zeta(\boldsymbol{\eta}_\ell)^2, \tag{69}$$

and the additional terms of the potential-energy density

$$\widehat{\mathcal{V}}_{[\eta\eta]\ell} = \sum_{\alpha\gamma\delta}\sigma^{\gamma\delta}_\ell\eta^{\alpha\gamma}_\ell\eta^{\alpha\delta}_\ell$$

$$= \frac{1}{2}\left(\underbrace{\mathrm{Tr}\{\sigma_\ell\}}_{=0}\boldsymbol{\eta}^2 - \sum_{\gamma\delta}\sigma^{\gamma\delta}_\ell\boldsymbol{\eta}_\gamma\boldsymbol{\eta}_\delta\right) \tag{70}$$

and

$$\widetilde{\mathcal{V}}_{[\varepsilon\eta]\ell} = \sum_{\alpha\gamma\delta}\sigma^{\gamma\delta}_\ell\eta^{\alpha\gamma}_\ell\varepsilon^{\alpha\delta} = \frac{1}{2}\boldsymbol{\tau}_\ell\cdot\boldsymbol{\eta}_\ell \tag{71}$$

with the coupling vector $\boldsymbol{\tau}_\ell$ defined in Eq. (65).

In the following it will be of use to write the components of this vector as linear combination of the strain components

$$\tau^\nu_\ell = \sum_{\alpha\beta}t^{(\nu),\alpha\beta}_\ell\varepsilon^{\alpha\beta} \tag{72}$$

with

$$t^{(\nu),\alpha\beta}_\ell = \frac{\partial}{\partial\varepsilon^{\alpha\beta}}\tau^\nu_\ell \tag{73}$$

The coefficients $t^{(\nu),\alpha\beta}_\ell$ are just combinations of local stresses. They are listed explicitly in Table 2.

The equation of motion (in frequency space with $z=\omega^2+i\varepsilon$), for $\boldsymbol{u}(\boldsymbol{R},z)$ is

$$-\rho_m z\,u^\alpha(\boldsymbol{R},z) = \sum_\beta\frac{\partial}{\partial x_\beta}\left(\sum_{\gamma\delta}C^{\alpha\gamma\delta\beta}(\boldsymbol{R})\varepsilon^{\gamma\delta}(\boldsymbol{R},z)\right.$$
$$\left.+\sum_\nu\sum_\ell s^{(\nu),\alpha\beta}_\ell\eta^\nu_\ell(\boldsymbol{R},z)\right) \tag{74}$$

with

$$s^{(\nu),\alpha\beta}_\ell = \frac{1}{2}\begin{cases} t^{(\nu),\alpha\alpha}_\ell & \alpha=\beta \\ \frac{1}{2}t^{(\nu),\alpha\beta}_\ell & \alpha\neq\beta \end{cases} \tag{75}$$

For $\boldsymbol{\eta}(\boldsymbol{R},z)$ we have

$$-\zeta z\,\eta^\nu_\ell(\boldsymbol{R},z) = \sum_\mu\sigma^{\nu\mu}_\ell\eta^\mu_\ell(\boldsymbol{R},z) - \tau^\mu_\ell(\boldsymbol{R},z) \tag{76}$$

Defining the non-coupled Green matrix of $\boldsymbol{\eta}(\boldsymbol{R},z)$ as

$$\left[\mathsf{G}_\ell(z)^{-1}\right]^{\nu\mu} = \sigma^{\nu\mu} + \zeta z\delta_{\nu\mu}, \tag{77}$$

we can solve for $\eta^\nu_\ell$:

$$\eta^\nu_\ell(\boldsymbol{R},z) = \sum_\mu\mathsf{G}^{\nu\mu}_\ell(z)\tau^\nu_\ell(\boldsymbol{R},z)$$
$$= \sum_\mu\mathsf{G}^{\nu\mu}_\ell(z)\sum_{\gamma\delta}t^{(\nu),\gamma\delta}_\ell\varepsilon^{\gamma\delta} \tag{78}$$

Inserting Eq. (78) into (74) we obtain

$$-\rho_m z\,u^\alpha(\boldsymbol{R},z)$$
$$= \sum_\beta\frac{\partial}{\partial x_\beta}\sum_{\gamma\delta}\left(C^{\alpha\gamma\delta\beta}(\boldsymbol{R}) + \Delta C^{\alpha\gamma\delta\beta}(z)\right)\varepsilon^{\gamma\delta}(\boldsymbol{R},z), \tag{79}$$

**Table 2 | List of the stress contributions $t^{(\nu),\alpha\beta}_\ell$**

| | | |
|---|---|---|
| $t^{(x),xx}=0$ | $t^{(y),xx}=-\sigma^{xz}$ | $t^{(z),xx}=\sigma^{xy}$ |
| $t^{(x),yy}=\sigma^{yz}$ | $t^{(y),yy}=0$ | $t^{(z),yy}=-\sigma^{xy}$ |
| $t^{(x),zz}=-\sigma^{yz}$ | $t^{(y),zz}=\sigma^{xz}$ | $t^{(z),zz}=0$ |
| $t^{(x),yz}=\sigma^{yy}-\sigma^{zz}$ | $t^{(y),yz}=-\sigma^{xy}$ | $t^{(z),yz}=-\sigma^{xz}$ |
| $t^{(x),xz}=-\sigma^{xy}$ | $t^{(y),xz}=\sigma^{xx}-\sigma^{zz}$ | $t^{(z),xz}=-\sigma^{yz}$ |
| $t^{(x),xy}=\sigma^{xz}$ | $t^{(y),xy}=-\sigma^{yz}$ | $t^{(z),xy}=\sigma^{yy}-\sigma^{xx}$ |

For brevity we omitted the subscript $\ell$ labelling the local regions.

where the additional contributions to the elastic coefficients are given by

$$\Delta C^{\alpha\gamma\delta\beta}(z) = \sum_{\nu\mu}\sum_{\ell} s_{\ell}^{(\nu),\alpha\beta} \, G_{\ell}^{\nu\mu}(z) \, t_{\ell}^{(\mu)\gamma\delta} \qquad (80)$$

We now simplify the treatment as follows:

Because all patches around a local stress $\sigma_{\ell}$ involve a vorticity pattern, which is just described by a single variable, namely $\boldsymbol{\eta}_{\ell}^{(\tilde{z})} \equiv \eta_{\ell}$, where $\tilde{z}$ indicates the $z$ axis of the local coordinate system pointing into the direction of $\boldsymbol{\eta}_{\ell}$, we work in terms of scalar variables $\eta_{\ell}$ with a Green's function

$$G_{\ell}(z) = \frac{1}{\zeta z - \sigma_{\ell}} \qquad (81)$$

where $\sigma_{\ell}$ is now any of the local stress components. Because the stress-induced correction to the elastic coefficients, given by Eq. (80) is just a quadratic form of the local stresses with the local Green's functions as coefficients we define – in the spirit of the isotropic approximation of the HET, Eq. (21)– the additional contribution to the frequency-dependent elastic self energy as

$$\Delta C^{\alpha\gamma\delta\beta}(z) \to \Sigma_{\varepsilon\eta}(z)\left(\delta_{\alpha\gamma}\delta_{\delta\beta} + \delta_{\alpha\delta}\delta_{\gamma\beta}\right) \qquad (82)$$

with

$$\Sigma_{\varepsilon\eta}(z) = \sum_{\ell} \sigma_{\ell}^2 G_{\ell}(z) \qquad (83)$$

where $\sigma_{\ell}$ is, again, any matrix element of the local stresses. The simplified version of the generalized heterogenous-elasticity (GHET) is now obtained by replacing

$$\Sigma(z) \quad \to \quad \Sigma_{\mathrm{HET}}(z) + \Sigma_{\varepsilon\eta}(z)$$

The contribution to the shear-modulus self energy takes the form

$$\Sigma_{\varepsilon\eta}(z) \sim \overline{\frac{\sigma_{\ell}^2}{-z\zeta + \sigma_{\ell}}} = \int d\sigma \mathcal{P}(\sigma)\frac{\sigma^2}{-z\zeta + \sigma} \qquad (84)$$

The overbar indicates the average over the local stresses and $\mathcal{P}(\sigma)$ is their distribution density.

The corresponding contribution to the spectrum $\rho_{\varepsilon\eta}(\lambda)$ is given by

$$\rho_{\varepsilon\eta}(\lambda) \sim \Sigma_{\varepsilon\eta}''(\lambda) \sim \sigma^2 \mathcal{P}(\sigma)\big|_{\sigma = \lambda\zeta} \qquad (85)$$

**Evaluation of $\mathcal{P}(\sigma)$**

In this appendix we discuss the general properties of the distribution of the local stress and what is expected for the low stress value behaviour of this distribution.

**Basic.** We are dealing with the local stress:

$$\sigma_{ij}^{\alpha\beta} \doteq r_{ij}^{\alpha} r_{ij}^{\beta} \frac{\phi'(r_{ij})}{r_{ij}} \qquad (86)$$

or, by introducing the unit vector $\hat{\mathbf{e}} \doteq \mathbf{r}/|\mathbf{r}|$

$$\sigma_{ij}^{\alpha\beta} = \hat{e}_{ij}^{\alpha}\hat{e}_{ij}^{\beta} \, r_{ij}\phi'(r_{ij}) \qquad (87)$$

For sake of simplicity, let's omit the subscript $ij$ and define $\psi(r) = r\phi'(r)$:

$$\sigma^{\alpha\beta} = \hat{e}^{\alpha}\hat{e}^{\beta} \, \psi(r) \qquad (88)$$

As discussed in paragraph M2, we are interested in the traceless stress tensor:

$$\tilde{\sigma}^{\alpha\beta} = \left(\hat{e}^{\alpha}\hat{e}^{\beta} - \frac{1}{d}\delta_{\alpha\beta}\right)\psi(r) \qquad (89)$$

The angular and radial part of the stress for a given couple of particle enter multiplicatively. Given the cylindrical symmetry of the problem (the angular part only depends on the orientation of the axis defined by the two particles) we expect that the angular part plays no role in the distribution if the system is isotropic. Indeed, we can make a rotation of the matrix $M^{\alpha\beta} = \hat{e}^{\alpha}\hat{e}^{\beta}$ so to orient the vector $\hat{\mathbf{e}}$ along the first coordinate. Being the matrix $M^{\alpha\beta}$ dyadic, its eigenvalues are all zero but one, which value is equal to one, and the corresponding eigenvector is parallel to $\hat{\mathbf{e}}$. After rotation, the stress matrix in any dimension is thus:

$$\overline{\overline{\sigma}} = \begin{pmatrix} 1 & 0 & \cdots & 0 \\ 0 & 0 & \cdots & 0 \\ \vdots & \vdots & \ddots & \vdots \\ 0 & 0 & \cdots & 0 \end{pmatrix}\psi(r)$$

and its traceless counterpart:

$$\overline{\overline{\tilde{\sigma}}} = \begin{pmatrix} \frac{d-1}{d} & 0 & \cdots & 0 \\ 0 & -\frac{1}{d} & \cdots & 0 \\ \vdots & \vdots & \ddots & \vdots \\ 0 & 0 & \cdots & -\frac{1}{d} \end{pmatrix}\psi(r)$$

As expected, the angular part is deterministic, and the distribution of the stress is controlled by the distribution of $\psi(r)$. The sign of the elements of the matrix $\tilde{\sigma}$ however can be both positive and negative. As for the inverse law potential investigated here $\psi(r)$ is always negative, the traceless stress matrix has $d-1$ positive and 1 negative elements. As discussed, only positive stress values give rise to non-phononic type II modes. Importantly, in any dimension, positive stress values do exist.

**Stress distribution.** The quantity that defines, for each couple of particles, the absolute value of the stress is $|\psi(r)|$. As it represents the value of the stress, let's call it '$\sigma$' (a scalar, not to be confused with the stress tensor). That is

$$\sigma \doteq |\psi(r)| = |r\phi'(r)| \qquad (90)$$

Let's now search for the distribution of $\sigma$, let's say $\mathcal{P}(\sigma)$.

The distribution of the $r_{ij}$ value is governed by the particle pair distribution function $g(r)$. Specifically in dimension $d = 2$:

$$P(r) = 2\pi\rho r^2 g(r) \qquad (91)$$

(where $\rho = N/\Omega$ is the particle density) and in $d = 3$:

$$P(r) = 4\pi\rho r^2 g(r) \qquad (92)$$

and, using

$$\mathcal{P}(\sigma)|d\sigma| = P(r)|dr| \qquad (93)$$

we get

$$\mathcal{P}(\sigma) = \sim r^{d-1} g(r) \left| \frac{d\sigma}{dr} \right|^{-1} \tag{94}$$

**The case of a smooth (tapered) cutoff.** In real numerical studies, the interaction potential is "shifted" and "tapered" in such a way to be zero at a finite $r$ value (at the cut-off value $r_c$) and to have continuity up to its m-th derivative at $r_c$. Its expression becomes:

$$
\begin{aligned}
\phi(r) &= a \left( r^{-n} - r_c^{-n} \right) \left( r_c^2 - r^2 \right)^m & r < r_c \\
\phi(r) &\sim (r_c - r)^{m+1} & r \lesssim r_c \\
\phi(r) &= 0 & r > r_c
\end{aligned}
\tag{95}
$$

The value of $m$ is (almost) always chosen to be $m = 2$, as the continuity of the first derivative of the potential (the force) is needed for the correct energy conservation during the dynamics, and the continuity of the second derivative guarantees a good estimation of the Hessian.

By defining $x = r/r_c$ we can rewrite the previous equation as:

$$
\begin{aligned}
\phi(x) &= a' \left( x^{-n} - 1 \right) \left( 1 - x^2 \right)^m & x < 1 \\
\phi(x) &= 0 & x > 1
\end{aligned}
\tag{96}
$$

Obviously $\sigma = |r\phi'(r)| = |x\phi'(x)|$.

Being interested in the $r$-region where the stress is small, we have now to consider the region around $r \approx r_c$, i.e. $x \approx 1$. In this case, it's worth to define $\epsilon = 1 - x^2$ and to expand the potential around $\epsilon = 0$ ($\epsilon > 0$ hereafter):

$$\phi(\epsilon) = a' \left( (1 - \epsilon)^{-n/2} - 1 \right) \epsilon^m \sim \epsilon^{m+1} \tag{97}$$

Now, rembering we are in the limit of small $\epsilon$:

$$
\begin{aligned}
\sigma = |x^2 \phi'(x)| = (1 - \epsilon) \left| \frac{d\phi(\epsilon)}{dx} \right| &\sim \left| \frac{d\epsilon^{m+1}}{dx} \right| \\
&= \frac{d\epsilon^{m+1}}{d\epsilon} \left| \frac{d\epsilon}{dx} \right| \sim \epsilon^m \left| \frac{d\epsilon}{dx} \right| = \epsilon^m \left| \frac{d(1 - x^2)}{dx} \right| \\
&= \epsilon^m x \sim \epsilon^m
\end{aligned}
\tag{98}
$$

Thus

$$\sigma = \epsilon^m \tag{99}$$

and

$$\epsilon = \sigma^{\frac{1}{m}} \tag{100}$$

Now

$$\frac{d\sigma}{dr} = \frac{d\sigma}{d\epsilon} \frac{d\epsilon}{dr} \sim \epsilon^{m-1}(-r) \approx \epsilon^{m-1}(-r_c) \tag{101}$$

thus

$$\left| \frac{d\sigma}{dr} \right| \sim \epsilon^{m-1} \sim \sigma^{\frac{m-1}{m}} \tag{102}$$

Finally, using Eq. (94), we have

$$\mathcal{P}(\sigma) \sim r_c^{d-1} g_2(r_c) \sigma^{\frac{1-m}{m}} \tag{103}$$

We observe that the behaviour of $\mathcal{P}(\sigma)$ for $\sigma \to 0$, does not depend on $n$. Rather it only depends on the value of $m$: it is the presence of a cut-off, and the continuity of $\phi(r)$ and of its first $m$ derivatives at the cut-off position that determine the stress distribution. Further, we

observe that for $m = 2$ we have $\mathcal{P}(\sigma) \sim \sigma^{-\frac{1}{2}}$, which, in turn, as we will see in the next paragraphs, implies $\rho(\omega) \sim \omega^4$.

**The case of a potential with a minimum.** If an interatomic potential $\phi(r)$ has a minimum, say, at $r = r_0$ like in the case of a Lennard-Jones (LJ) potential, then the first derivative passes *linearly* through zero at $r = r_0$. Obviously, this corresponds to the case of a tapered potential with $m = 1$ Correspondingly we have

$$\mathcal{P}(\sigma) \sim r_0^{d-1} g_2(r_0) \sigma^0 = const. \tag{104}$$

It is plausible that the overall distribution of the small stresses is dominated by this contribution because $g_2(r)$ is maximal at $r_0$.

**From $\mathcal{P}(\sigma)$ to $g(\omega)$.** As discussed in the main text and in M2, there are a direct and an indirect contributions of the type-II excitations to the spectrum, $\rho_\eta(\lambda)$ and $\rho_{\varepsilon\eta}(\lambda)$, given by ($\lambda = \omega^2$)

$$\rho_{\varepsilon\eta}(\lambda) = \frac{1}{2\omega} g(\omega) \sim \sigma^2 P(\sigma) \big|_{\sigma = \lambda\zeta} \tag{105}$$

From this, we obtain the following relations for the spectra

<u>*Potential with cutoff and tapering*</u>

$$
\begin{aligned}
\rho_{\varepsilon\eta}(\lambda) &\sim \lambda^{\frac{1}{m}+1} \\
g_{\varepsilon\eta}(\omega) &\sim \omega^{\frac{2}{m}+3}
\end{aligned}
\tag{106}
$$

<u>*Potential with minimum at $r = r_0$*</u>

$$
\begin{aligned}
\rho_{\varepsilon\eta}(\lambda) &\sim \lambda^2 \\
g_{\varepsilon\eta}(\omega) &\sim \omega^5
\end{aligned}
\tag{107}
$$

## Simulation details

We performed numerical simulations using the MC-swap algorithm of a $d$-dimensional ($d = 3$ here) system composed of $N = 10^d$ particles in a box (with periodic boundary conditions) of side $L = N^{1/d}$ so that the number density is $\rho = N/L^d = 1$.

To find the inherent glass structures, we consider Monte Carlo (MC) dynamics combined with swap moves for equilibrating the system in a wide range of temperatures. We compute stable glass configurations from different parent temperatures by minimizing the configurational energy $\Phi$ and thus obtaining the corresponding inherent structures through the Limited-memory Broyden-Fletcher-Goldfarb-Shanno (LBFGS) algorithm[81]. The spectrum of the harmonic oscillations around the inherent configurations has been obtained by computing all the $dN$ eigenvalues of the hessian matrix using Python NumPy linear algebra functions[82]. We produced data for two tapering functions $m = 2$ and $m = \infty$. The potential is given in Eq. (14) of the main text.

Following early works on MC-swap[70,71], at each MC step we perform a swap move instead of a standard MC step with probability $p_{swap} = 0.2$. The displacement vector $\Delta\mathbf{r}$ of the standard MC move has modulus $|\Delta\mathbf{r}| = 0.2$ so that the acceptance probability falls into the interval 30% (low temperature) to 60% at high temperatures. We consider runs of $N_t = 1.1 \times 10^6$ MC steps. The power of the repulsive potential is $n = 12$.

## Data availability

All data shown in Figs. 2–4 are available on request from the authors.

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

## Acknowledgements

We are grateful to Hideyuki Mizuno and Vishnu Krishnan for discussions concerning small Lennard-Jones systems. M.P. has received funding from the European Union's Horizon 2020 research and innovation programme under the MSCA grant agreement No. 801370 and by the Secretary of Universities and Research of the Government of Catalonia through Beatriu de Pinós program Grant No. BP 00088 (2018). G.S. acknowledges the support of NSF Grant No. CHE 2154241.

## Author contributions

G.R. and W.S. set up the project and worked out the theory. M.P., F.C.M., and D.K. performed the simulations, G.S. and F.Z. supported the analysis, G.R. and W.S. wrote the manuscript with contributions from all authors.

## Competing interests

The authors declare no competing interests.
