## [Peer Review File · Nature Communications]

The nature of non-phononic excitations in disordered systemsREVIEWER COMMENTS

Reviewer #1 (Remarks to the Author):

This work investigates an important problem in glass physics and presents interesting progress on the understanding of the vibrational modes in disordered solids. The non-phononic modes are classified into two types and their properties dependence on the system size and interatomic potential are revealed. It is surprising to see that the cutoff in the potential can have such a huge influence on the vibrational properties, which should be common to computer simulations of disordered states. This study possibly brings an end to the chaos era on the scaling property of the low-frequency modes. Therefore, I would like to recommend its timely publication pending the following concerns/suggestions addressed.

Even though with heavy theoretical discussion, it is not so clear the nature difference between type-I and type-II non-phononic modes. If these modes are related to the “dynamical defects” in glasses, they are naturally explained on their dependence on the glass's stability. Why such classification is required? From the atomic-scale features, how do these two types differ from each other?

As for a research article in Nat. Commun., I would suggest the authors make the theoretical part more concise and put more details in the supplementary information for a broader range of readership.

If type-II non-phononic modes originate from finite-size effect, while boson peak do not, it deserves further discussion on the relationship between the former and the latter. In the literature, the four-leaf pattern quasi-localized modes have been assumed to contribute to the boson peak.

From the definition of type-I and type-II non-phononic modes, how do the authors explain the asymmetric feature of the reduced DOS?

In Figure 4, as always, the problem is the fitting frequency range is so small for a power-law function, how would the authors convince the readers that the exponent is robust?

From the heterogeneous elasticity theory, how can one connect the frequency range of the non-phononic modes to the length scale of HET or GHET? More quantitative evidence from either simulations or experiments is required.

What is the role of anharmonicity in determining these two types of non-phononic modes?

Reviewer #2 (Remarks to the Author):

In their manuscript titled "The Nature of Non-phononic Excitations in Disordered Systems," authored by Schirmacher et al., a comprehensive analysis of the low-frequency vibration density of states in disordered materials is presented. This analysis combines both analytical theories and numerical simulations to shed light on the subject. The scientific problem's setup is succinctly summarized in Fig. 1 of the manuscript.

In the context of real experimental measurements on a molecular glass, where the system size approaches the thermodynamical limit, the density of states is found to be influenced by

whether the system behaves as a stable glass or a marginal glass. This distinction is clearly illustrated in panels a and b of Fig. 1. In numerical simulations, differences arise due to the finite system size, introducing new features related to the low-frequency vibration density of states, as depicted in panel c of Fig. 1.

Concerning the circumstances in Fig. 1c, the authors have developed a novel analytical field theoretical model known as the Generalized Heterogeneous-Elasticity Theory (GHET). This model establishes a connection between the scaling exponent of the low-frequency density of states and the tapering function used in numerical studies to ensure the first m derivatives of the interaction potential are zero. The manuscript succinctly summarizes the main theoretical results in equations 8-10, and these findings are further supported by their numerical studies.

The results presented in this manuscript are thought-provoking and potentially have significant implications for our understanding of the origin of the low-frequency vibration density of states in small systems. Indeed, this has been a focal point of extensive numerical investigations in recent years.

However, I still have two primary concerns that I hope the authors can address before I can make a decision regarding the manuscript:

In the GHET theory, the current formulation lacks an explicit dependence on the system size. As elucidated in Fig. 1c, this explicit consideration of system size is of paramount importance. It would significantly enhance the clarity of the theoretical predictions if an explicit system size parameter were incorporated into the theory. Alternatively, the authors might consider providing a colloquium discussion that intricately links the GHET theory to the system size dependence if incorporating such a parameter proves to be challenging.

My second concern pertains to the numerical studies in the present work. As noted by the authors in Fig. 1c, the low-frequency scaling of the density of states in stable glasses is attributed to the finite system size. To bolster the credibility of the results, it would be advantageous to conduct a finite-size analysis in the numerical studies. This analysis can demonstrate how the asymptotic behavior is derived by extrapolating numerical results from various system sizes.

Addressing these concerns would significantly strengthen the manuscript and its contributions to the field.

Reviewer #3 (Remarks to the Author):

In this study, the authors generalize the heterogeneous-elasticity theory by including the effects of local non-irrotational oscillations associated with the stress field. The main outcome is a theoretical prediction of the low-frequency scaling of the density of non-phononic vibrational states (called type II). According to the theory, the exponent of this scaling depends on the statistics of the small values of the local stresses. This prediction is examined by numerical simulations of glass models.

The significance of the study, as stated by the title, is to reveal the nature of non-phononic excitations in glasses (amorphous solids). Glasses have very different vibrational properties

compared to crystals. For example, their vibrational density of states (vDOS) does not follow Debye's law in the low-frequency regime, and displays a boson peak. In recent years, a quartic low-frequency scaling of the vDOS, $g(\omega) \sim \omega^4$, has been observed in many simulations. It is a fundamental task to understand the nature of the vibrational behavior of amorphous solids, in order to develop a solid-state physics theory of such materials. The present study, as a first-principle theory, has its value towards this goal. I believe that the manuscript should be published ultimately. Before that, however, I suggest the authors to clarify the following questions.

1) In the introduction, can the authors provide clear and explicit definitions of type-I and type-II non-phononic modes?

2) In many simulation studies, for example, Ref. [41], a short-range, purely repulsive, harmonic potential is used. For this model, the potential vanishes at r_c (the particle diameter) with a power law $(r_c - r)^2$, which means that $m = 1$ in Eq. (12). In such simulations, the quartic scaling $g(\omega) \sim \omega^4$ is observed [41]. However, the theoretical result Eq. (12) predicts $g(\omega) \sim \omega^5$ for $m = 1$. Can this discrepancy between the theory and the simulation result be explained? Many simulation studies focus on harmonic and Hertzian potentials as models of granular matter (a kind of athermal glass). Besides the tapered potential Eq. (10) and the LJ potential that have already been considered, it is worth discussing in detail the theoretical predictions for such granular models and comparing them to reported simulation results.

3) The $g(\omega) \sim \omega^4$ behavior has been attributed to the anharmonic effect in the interaction potential [13]. For example, this scaling can be derived from a simple soft-potential model, which adds a quartic term to the expression of the potential [13]. What is the advantage of the current theory compared to the soft-potential model? In the present approach, in order to obtain type II modes, it is essential to include a non-irrotational vector field, which "is similar to that of a set of local oscillators coupled to the strain field". Do such local oscillators effectively introduce an anharmonic effect to the potential? Can the current approach be reconciled with the much simpler soft-potential model in this sense?

4) On Page 6, "The states in this region are known [14, 22] to obey the statistics of random-matrix eigenstates." This sentence needs to be rephrased. The origin of the boson peak is still under debate; see, e.g., [Hu & Tanaka, Nature Physics 18.6 (2022): 669-677] for a different viewpoint. The two references [14,22] are theoretical studies. It remains unclear whether the states near the boson peak obey the statistics of random-matrix eigenstate in experimental systems. If such experimental verification exists, it should be cited and discussed. In fact, one can argue that the random-matrix theory is over-simplified because it completely neglects the correlations in the interaction (contact) network -- it assumes that the network is mean-field, tree-like [22]. In contrast, the real network has a complex structure with abundant loops. It would be useful if the authors could discuss and review other viewpoints on the origin of the boson peak.

5) Fig. 2 shows that type I and type II modes have universal level distance statistics. The authors conclude that both types are extended and not localized, meaning that they can not be differentiated by the participation ratio. Then, how does one distinguish between type I and type II modes at the single-mode level, apart from the spectrum? The difference between phonons and non-phonons is straightforward to see (even by eye) once plotted: phonons are wave-like, while non-phonons are disordered. If one plots a typical type I mode and a typical type II mode, how do they differ? If they cannot be distinguished at the single-

mode level, one should reconsider calling them two different “types” -- maybe there is only one type of non-phononic states, which, however, appear with varying densities in different frequency regimes. It is important to understand the nature and properties of non-phononic excitations, not only in the spectrum, but also in the real space.

6) One of the main theoretical predictions is the correspondence between the power-law scaling of the small local stress $P(\sigma) \sim \sigma^{-1+1/m}$ and the power-law scaling of the low-frequency modes $g(\omega) \sim \omega^{3+2/m}$. This prediction is supported by the simulation data in Fig. 3 and Fig. 4. However, the difference in the exponent ($s=3$ and 4) is not very big (Fig. 4B). It would be useful to examine $P(\sigma) \sim \sigma^{-1+1/m}$ and $g(\omega) \sim \omega^{3+2/m}$ in simulation models with $m=1$ and correspondingly $s=5$ (the third column in Table I)? This would enlarge the difference in the exponent (from $s=3$ to $s=5$) and make the comparison between the theory and simulations more convincing.

NCOMMS-23-47627-T

W. Schirmacher et al. :

“The nature of non-phononic excitations in disordered systems”

Answers to the Reviewers

We are grateful to the Reviewers for estimating our work worth being published, for their thorough criticism, and for very helpful advice.

The remarks of the Reviewers are printed in **blue**, our comments in **black**, and the revised text in **red**.

Reviewer # 1:

This work investigates an important problem in glass physics and presents interesting progress on the understanding of the vibrational modes in disordered solids. The non-phononic modes are classified into two types and their properties dependence on the system size and interatomic potential are revealed. It is surprising to see that the cutoff in the potential can have such a huge influence on the vibrational properties, which should be common to computer simulations of disordered states. This study possibly brings an end to the chaos era on the scaling property of the low-frequency modes. Therefore, I would like to recommend its timely publication pending the following concerns/suggestions addressed.

Even though with heavy theoretical discussion, it is not so clear the nature difference between type-I and type-II non-phononic modes. If these modes are related to the “dynamical defects” in glasses, they are naturally explained on their dependence on the glass's stability. Why such classification is required? From the atomic-scale features, how do these two types differ from each other?

The type-I and type-II excitations are distinguished by the nature of the vibrational wave functions. While the type-I modes are governed by strains in the presence of spatially fluctuating elastic moduli, the type-II modes are circular vortex-like modes around microscopic frozen-in stresses. In our opinion the type-I modes are those responsible for the appearance of the boson peak in very large systems. Local stresses have not been discussed in the context of non-phononic modes. Therefore we introduced the classification into type I and II. In order to make the distinction between the two types of non-phononic modes clearer, we modified one sentence in the introduction as follows:

In contrast to the type-I modes, which are caused by spatially fluctuating elastic coefficients [19, 20, 33] and which involve strains, i.e. irrotational displacement fields, the type-II modes are associated with local, frozen-in stresses and involve non-irrotational, vortex-like displacement patterns.

As for a research article in Nat. Commun., I would suggest the authors make the theoretical part more concise and put more details in the supplementary information for a broader range of readership.

We re-wrote the theoretical part and shortened it considerably. We provide the previous long version as supplementary material.

If type-II non-phononic modes originate from finite-size effect, while boson peak do not, it deserves further discussion on the relationship between the former and the latter.

The type-II modes are *not* due to a finite-size effect. They exist in samples of all sizes. They just become visible in the low-frequency regime of simulated spectra of systems with small size. In such small-size systems the waves (“phonons”), which in large systems dominate the low-frequency spectrum are absent. Also the boson peak is affected by the size of the system. In small-size systems the boson peak is *absent*, because the boson peak is the crossover from Debye’s wave spectrum to a random-matrix-like spectrum. In small systems, low-frequency Debye-wave do not exist. Instead of the boson peak a cross-over from type-II to type-I modes occurs.

We changed the text on type-I modes and the boson peak as follows:

[In large samples] ... the square-root in Eq. (2) produces its own imaginary part, leading to a shoulder in the spectrum. This shoulder appears as a maximum in the “reduced DOS” $g(\omega)/\omega^2$, and has been called “boson peak” for historical reasons [33]. So the boson peak is the crossover from a Debye-spectrum to a random-matrix spectrum.

[In small samples] ... in the absence of phonons, G_0 has still a finite value, but the low-frequency imaginary part $G''(\lambda)$ is gone. Instead of a boson peak Eq. (2) predicts now a gap in the spectrum.

We changed the text on type-II modes as follows

While the features of the strains are basically controlled by the space dependence of the fluctuating elastic constants and give rise to the type-I non-phononic modes (predicted by “standard” HET), the vorticities are associated with spatially fluctuating local stresses..

In the literature, the four-leaf pattern quasi-localized modes have been assumed to contribute to the boson peak.

The term quasi-localized modes has been coined by Laird and Schober [23] and is used nowadays for non-phononic modes. As a matter of fact, four-leaf displacement patterns appear at the hyperbolic point between four type-II vortices of opposite rotational direction. At present we cannot say, how much type-II stress-related vortex-like modes contribute to the boson peak in the presence of waves in large systems. This subject to future research. We inserted the following sentence into the conclusion section:

How the local stresses and the associated type-II modes enter into the vibrational spectrum of macroscopic samples, in particular, how they may influence the boson peak, will be a matter of future research.

From the definition of type-I and type-II non-phononic modes, how do the authors explain the asymmetric feature of the reduced DOS?

We are not very sure which kind of systems the Reviewer refers to. In spectra of simulated large glassy systems and in experimentally measured spectra of glasses the reduced DOS (boson peak) is asymmetric and can be satisfactorily explained by fluctuating elastic constants (HET, Type I). In small systems - as there is no boson peak due to a lack of waves - one usually does not look at the reduced DOS. Anyway there is no reason, why the reduced DOS of all of these systems should be symmetric.

In Figure 4, as always, the problem is the fitting frequency range is so small for a power-law function, how would the authors convince the readers that the exponent is robust?

For the time being, the most important results of our calculations are not the robustness of the exponents, but rather, that for high parental temperature the DOS does not change with the change of tapering, but for low parental temperature it does. We are pleased that, furthermore, the exponent, which can be extracted from the plots agree to the predictions of HET and GHET. In order to make this clear, we replaced the paragraph before the conclusions by

In other words: The main result of Fig. 4 is the *insensitivity* of the DOS on the tapering exponent m for high parental temperature T^* , as opposed to a sensitivity on m for low T^* . The observed low-frequency exponents compare well with the predictions of HET and GHET: $s = 2$ for high T^* , signifying type-I excitations at marginality, and $s = 4, 3$ for low T^* , corresponding to type-II excitations, which depend on the stress distributions for tapered potentials with $m = 2, \infty$.

From the heterogeneous elasticity theory, how can one connect the frequency range of the non-phononic modes to the length scale of HET or GHET?

The frequency range of the non-phononic modes is not related to a length scale in principle. On the other hand, elasticity theory, and therefore HET, is based on a coarse-graining procedure with a coarse-graining volume of mesoscopic size, whereas the stresses, which cause the type-II excitations are supposed of microscopic size. For high parental temperature, we think the system to be at marginal stability, and the type-I excitations to be dominant, there is not a particular length scale involved, as opposed to the type-II modes around the microscopic stresses. The low-frequency spectrum of these modes is only visible in systems small enough not to support standing waves. We included this discussion already in the previous version of the manuscript. However, to make the length-scale issue clearer, we inserted the following sentence:

At this point we note that that the local stresses, which give rise to the type-II modes involve a length scale of a few interatomic distance, whereas the heterogeneous elasticity, responsible for the type-I modes involves the mesoscopic length scale of the coarse-graining procedure.

More quantitative evidence from either simulations or experiments is required.

We agree to this. However, we feel that the further investigations of the two types of excitations is a very interesting challenge for future research. Concerning the future research we inserted the following paragraph into the Conclusions:

To investigate the difference of the two types of non-phononic excitations in more detail is an interesting task for future research. Similarly, it will be very interesting to investigate, how the local stresses and the associated type-II modes enter into the vibrational spectrum of macroscopic samples, and, in particular, how they may influence the boson peak.

What is the role of anharmonicity in determining these two types of non-phononic modes?

In the present paper we do not consider anharmonic vibrational phenomena at all. The present theory refers to the Hessian of a glass at zero temperature, which is the coefficient matrix of the harmonic part of the potential energy of the system. Anharmonic effects, which dominate the experimentally measured spectra at elevated temperatures and at frequencies much below the boson peak (in the GHz range) have been addressed previously in the literature. The influence of frozen-in stresses on anharmonic features is beyond the scope of the present paper. We modified the abstract and the introduction to make clear that we only treat a harmonic system.

Reviewer #2:

In their manuscript titled "The Nature of Non-phononic Excitations in Disordered Systems," authored by Schirmacher et al., a comprehensive analysis of the low-frequency vibration density of states in disordered materials is presented. This analysis combines both analytical theories and numerical simulations to shed light on the subject. The scientific problem's setup is succinctly summarized in Fig. 1 of the manuscript.

In the context of real experimental measurements on a molecular glass, where the system size approaches the thermodynamical limit, the density of states is found to be influenced by whether the system behaves as a stable glass or a marginal glass. This distinction is clearly illustrated in panels a and b of Fig. 1. In numerical simulations, differences arise due to the finite system size, introducing new features related to the low-frequency vibration density of states, as depicted in panel c of Fig. 1.

Concerning the circumstances in Fig. 1c, the authors have developed a novel analytical field theoretical model known as the Generalized Heterogeneous-Elasticity Theory (GHET). This model establishes a connection between the scaling exponent of the low-frequency density of states and the tapering function used in numerical studies to ensure the first m derivatives of the interaction potential are zero. The manuscript succinctly summarizes the main theoretical results in equations 8-10, and these findings are further supported by their numerical studies.

The results presented in this manuscript are thought-provoking and potentially have significant implications for our understanding of the origin of the low-frequency vibration density of states in small systems. Indeed, this has been a focal point of extensive numerical investigations in recent years.

However, I still have two primary concerns that I hope the authors can address before I can make a decision regarding the manuscript:

In the GHET theory, the current formulation lacks an explicit dependence on the system size. As elucidated in Fig. 1c, this explicit consideration of system size is of paramount importance. It would significantly enhance the clarity of the theoretical predictions if an explicit system size parameter were incorporated into the theory. Alternatively, the authors might consider providing a colloquium discussion that intricately links the GHET theory to the system size dependence if incorporating such a parameter proves to be challenging.

There is in principle no system-size dependence of type-II non-irrotational vibrational excitations as described by GHET. The system size comes into play if one considers the presence or absence of waves in the system. In small samples the lowest-frequency waves are standing waves, which show up as a peak in the spectrum and have a frequency $\omega_0 = 2\pi v_T / L$, where L is the system size and v_T the transverse sound velocity.

We inserted the following sentences into the paragraph, which deals with panel c) of Fig. !:

[... lowest resonance frequency ...] $\omega_0 = 2\pi v_T / L$, where v_T is the transverse sound velocity. As sketched in the Figure, ω_0 gives the upper limit for the visibility of the type-II spectrum. At higher frequencies these modes hybridize with the waves and probably can no more be distinguished from them.

My second concern pertains to the numerical studies in the present work. As noted by the authors in Fig. 1c, the low-frequency scaling of the density of states in stable glasses is attributed to the finite system size. To bolster the credibility of the results, it would be advantageous to conduct a finite-size analysis in the numerical studies. This analysis can demonstrate how the asymptotic behavior is derived by extrapolating numerical results from various system sizes. Addressing these concerns would significantly strengthen the manuscript and its contributions to the field.

In both Type-I and Type-II modes there exists in principle no scaling with the system size. We emphasize again that the system size comes only into play via the standing waves, the frequency of which scale with the inverse of the system size. For large system sizes these discrete-frequency modes lose importance, and the type-II modes (which possibly hybridize with the waves) are no more visible in the spectrum. We do not see any sense in performing a finite-size scaling analysis. However - also in response to concerns of Reviewer 1 - we inserted the following remark on length scales:

At this point we note that that the local stresses, which give rise to the type-II modes involve a length scale of a few interatomic distance, whereas the heterogeneous elasticity, responsible for the type-I modes involves the mesoscopic length scale of the coarse-graining procedure.

Reviewer #3 (Remarks to the Author):

In this study, the authors generalize the heterogeneous-elasticity theory by including the effects of local non-irrotational oscillations associated with the stress field. The main outcome is a theoretical prediction of the low-frequency scaling of the density of non-phononic vibrational states (called type II). According to the theory, the exponent of this scaling depends on the statistics of the small values of the local stresses. This prediction is examined by numerical simulations of glass models.

The significance of the study, as stated by the title, is to reveal the nature of non-phononic excitations in glasses (amorphous solids). Glasses have very different vibrational properties compared to crystals. For example, their vibrational density of states (ν DOS) does not follow Debye's law in the low-frequency regime, and displays a boson peak. In recent years, a quartic low-frequency scaling of the ν DOS, $g(\omega) \sim \omega^4$, has been observed in many simulations. It is a fundamental task to understand the nature of the vibrational behavior of amorphous solids, in order to develop a solid-state physics theory of such materials. The present study, as a first-principle theory, has its value towards this goal. I believe that the manuscript should be published ultimately. Before that, however, I suggest the authors to clarify the following questions.

1) In the introduction, can the authors provide clear and explicit definitions of type-I and type-II non-phononic modes?

In the introduction, we now characterize the two types of non-phononic modes as follows:

In contrast to the type-I modes, which are caused by spatially fluctuating elastic coefficients [19, 20, 33] and which involve strains, i.e. irrotational displacement fields, the type-II modes are associated with local, frozen-in stresses and involve non-irrotational, vortex-like displacement patterns.

Further, in the beginning of the paragraph on type-II modes we inserted:

While the features of the strains are basically controlled by the space dependence of the fluctuating elastic constants and give rise to the type-I non-phononic modes (predicted by "standard" HET), the

vorticities are associated with spatially fluctuating local stresses ...

2) In many simulation studies, for example, Ref. [41], a short-range, purely repulsive, harmonic potential is used. For this model, the potential vanishes at r_c (the particle diameter) with a power law $(r_c - r)^2$, which means that $m = 1$ in Eq. (12). In such simulations, the quartic scaling $g(\omega) \sim \omega^4$ is observed [41]. However, the theoretical result Eq. (12) predicts $g(\omega) \sim \omega^5$ for $m = 1$. Can this discrepancy between the theory and the simulation result be explained? Many simulation studies focus on harmonic and Hertzian potentials as models of granular matter (a kind of athermal glass). Besides the tapered potential Eq. (10) and the LJ potential that have already been considered, it is worth discussing in detail the theoretical predictions for such granular models and comparing them to reported simulation results.

We have deliberately not considered tapered $m=1$ potentials, because their Hessian is discontinuous. Consequently, numerical simulations based on such potentials tend to be unstable, and their results of limited significance. Further, the simulations presented in Ref. [41] concern a rather large system in which low-frequency waves (phonons) are present. There is no discrepancy to the theoretical result of Eq. (12), because this result is based on the absence of phonons, while the system studied in Ref. [41] contained low-frequency phonons. We are not convinced by the procedure followed in Ref. [41] separating the phonon and the non-phonon spectra.

We inserted the following sentence into the discussion of the tapering exponent m :

There exists a large body of simulations with with a potential, which quadratically becomes zero at a radius r_c (see Ref. [41] for references). Such potentials have been used in jamming studies [70, 71]. Because the Hessian of this potential is discontinuous at $r = r_c$, and due to the mentioned imponderabilities of simulations with $m < 2$, we exclude these systems from the present discussion.

3) The $g(\omega) \sim \omega^4$ behavior has been attributed to the anharmonic effect in the interaction potential [13]. For example, this scaling can be derived from a simple soft-potential model, which adds a quartic term to the expression of the potential [13]. What is the advantage of the current theory compared to the soft-potential model?

The present theory concerns exclusively harmonic vibrations. It is based on the harmonic term of the potential energy of the system. The theory provides a microscopic derivation of the equations of motions from the system's Hessian. The soft-potential model invokes the anharmonic interaction explicitly. Our work shows that anharmonicity is not needed in order to explain the vibrational anomalies of glasses.

We modified the abstract and the introduction to make clear that we only consider harmonic systems.

In the present approach, in order to obtain type II modes, it is essential to include a non-irrotational vector field, which "is similar to that of a set of local oscillators coupled to the strain field". Do such local oscillators effectively introduce an anharmonic effect to the potential? Can the current approach be reconciled with the much simpler soft-potential model in this sense?

If one considers the defect states of the soft-potential model as local oscillators, which are linearly coupled to the waves (as is assumed in soft-potential papers on sound attenuation) one may construct such an analogy. However, we think that our approach is much simpler (in the sense of Occam's Razor) than the soft-potential model, because it does not invoke the anharmonic interaction and just takes the system's Hessian as starting point. Further, in $T=0$ simulations the

anharmonic interaction should not play a significant role. The observed ω^4 scaling of the non-phononic excitations, as we show, is not due to anharmonic defects but due to the $m=2$ tapering.

4) on page 6, “the states in this region are known [14, 22] to obey the statistics of random-matrix eigenstates.” This sentence needs to be rephrased. The origin of the boson peak is still under debate; see, e.g., [Hu & Tanaka, Nature Physics 18.6 (2022): 669-677] for a different viewpoint. The two references [14,22] are theoretical studies. It remains unclear whether the states near the boson peak obey the statistics of random-matrix eigenstate in experimental systems. If such experimental verification exists, it should be cited and discussed. In fact, one can argue that the random-matrix theory is over-simplified because it completely neglects the correlations in the interaction (contact) network -- it assumes that the network is mean-field, tree-like [22]. In contrast, the real network has a complex structure with abundant loops.

We agree that the phrase is misleading. We modified it as

It has been suggested [14, 22] that the states in this region obey the statistics of random-matrix eigenstates.

It would be useful if the authors could discuss and review other viewpoints on the origin of the boson peak.

In the last 30 years a very large number of viewpoints on the origin of the boson peak have been published. As the emphasis of the present study is not a discussion of the boson peak, we refrain from discussing all these viewpoints. We now refer to the different views of Chumakov et al. and Tanaka et al. and then refer to a recent review published by two of the present authors, in which a large number of proposed models for the boson peak are discussed. The new text in the introduction is now:

The nature of the boson- peak anomaly has been debated controversially [31, 32] (see [33] for a listing and a discussion of various proposed models for the the boson peak), but - as we feel - in the light of heterogeneous-elasticity theory (HET) [19, 20, 34] it became clear ...

5) Fig. 2 shows that type I and type II modes have universal level distance statistics. The authors conclude that both types are extended and not localized, meaning that they can not be differentiated by the participation ratio. Then, how does one distinguish between type I and type II modes at the single-mode level, apart from the spectrum? The difference between phonons and non-phonons is straightforward to see (even by eye) once plotted: phonons are wave-like, while non-phonons are disordered. If one plots a typical type I mode and a typical type II mode, how do they differ? If they cannot be distinguished at the single-mode level, one should reconsider calling them two different “types” -- maybe there is only one type of non-phononic states, which, however, appear with varying densities in different frequency regimes. It is important to understand the nature and properties of non-phononic excitations, not only in the spectrum, but also in the real space.

In the already existing text of the manuscript we tried to make clear that the prominent features of the Type-II modes, which make them qualitatively different from the Type-I modes, (which are composed of irrotational displacements), is their non-irrotational, i.e. vortex-like character. In order to convince the referee that vortices are present in the wavefunctions of the Type-II excitations, we show here in the Answers to the Reviewers a picture of a slice of a Type-II wavefunction, obtained by quenching from a low parental temperature. We refrained from including such a figure in the text, because a proper investigation of the non-irrotational character of the Type-II wave functions would require a pertinent statistical analysis, which, to our opinion, is out of the scope of this first

paper on the stress-induced vibrational Type-II excitations. However, we believe that such an investigation is important and necessary and will be performed in the near future.

FIG. 1. Example of a real-space representation of a Type-II eigenmode. We show a slice (thickness σ) of the three-dimensional eigenmode of the 12th eigenvalue corresponding to the $m = 2$ spectra of panel (B) of Fig. 4, projected into the $(x - y)$ plane of the slice. Lengths of the arrows indicate the mode amplitude.

6) One of the main theoretical predictions is the correspondence between the power-law scaling of the small local stress $P(\sigma) \sim \sigma^{-1+1/m}$ and the power-law scaling of the low-frequency modes $g(\omega) \sim \omega^{3+2/m}$. This prediction is supported by the simulation data in Fig. 3 and Fig. 4. However, the difference in the exponent ($s=3$ and 4) is not very big (Fig. 4B). It would be useful to examine $P(\sigma) \sim \sigma^{-1+1/m}$ and $g(\omega) \sim \omega^{3+2/m}$ in simulation models with $m=1$ and correspondingly $s=5$ (the third column in Table I)? This would enlarge the difference in the exponent (from $s=3$ to $s=5$) and make the comparison between the theory and simulations more convincing.

We agree that comparing with data obtained with $m = 1$ tapering (if reliable) would make a clearer difference. However we indicated already our concerns on systems with $m=1$, which precludes taking such data into account.

REVIEWER COMMENTS

Reviewer #1 (Remarks to the Author):

The authors have made a great effort to address my concerns. I still think this work makes a great contribution to advancing our understanding of the low-frequency modes in glassy solids. I would like to recommend its publication in Nature Communications.

Reviewer #2 (Remarks to the Author):

I have meticulously reviewed the revised manuscript prepared by Walter Schirmacher and colleagues, taking note of the thorough responses to concerns raised by all three referees. The authors have demonstrated a commendable commitment to addressing feedback, resulting in substantial revisions that notably enhance the quality and clarity of their work.

The presented results and theories in this manuscript are both thought-provoking and hold significant implications for our understanding of the low-frequency vibration density of states in small systems. The manuscript has transformed into an exemplary piece, and I wholeheartedly recommend its publication in Nature Communications.

Reviewer #3 (Remarks to the Author):

Question (2): in my previous report, I asked if the authors can use their theoretical model to explain the ω^4 scaling observed previously in simulations of granular models. The reason is that a naive implementation of their Eq. (12) gives an inconsistent ω^5 scaling. This means that their theory is in contradiction with simulations in such models.

I find the answer by the authors unsatisfactory. They exclude such models in their considerations, which means that the applicability of their theory is limited. More importantly, this makes me doubt how universal the proposed mechanism is, if it can not be used to explain granular models where the low-frequency ω^4 scaling has been repeatedly reported.

They further argue that the results in Ref. [41] are of limited significance because of the presence of phonons. However, Ref. [41] is not the only study that reports ω^4 scaling in a one-sided harmonic potential (HARM). For example, in Ref [38] by Lerner et al., small systems are simulated where the phonons are absent, and the ω^4 scaling is also observed in the HARM model. In fact, in Ref.[38], the authors simulated three different models, and the low-frequency scaling looks rather universal according to their data. Thus the scaling seems independent of whether the derivative of the potential is continuous or discontinuous.

Question (3): I asked whether the current theory has a clear advantage compared to previous theories, such as the soft-potential model that considers the anharmonic effect.

The answer is also unsatisfactory. The authors emphasize that they do not consider the anharmonic effect, and argue that the anharmonicity is not needed to explain the vibrational

anomalies of glasses. On the other hand, their theory cannot explain the behavior of models whose Hessian is discontinuous, as discussed above. Thus theory theory does not work universally.

In short, I cannot recommend for publication unless the authors can improve the current theory to explain the scaling observed in granular models. I cannot exclude the possibility that the current mechanism only works for specific systems. I do not doubt that the exponent indeed depends on the interaction potential in the setup considered in the current study, but perhaps this effect is not that important in general situations. In particular, there are alternative mechanisms to explain the low-frequency vibrational scaling (e.g., the anharmonic effect). Perhaps other mechanisms dominate in the general case, and that is why the ω^4 scaling is universally observed in models (e.g., the HARM model) to which the current theory fails to apply.

Point-to-point reply to the Referee's concerns

Here we summarize the two concerns of Reviewer #3:

(2) The Reviewer raises doubts concerning the generality of our treatment, because we excluded one-sided harmonic potentials with discontinuous Hessian matrix elements (Refs. [41] and [38]) from the discussion. He further claims that the results of these studies are in conflict with our predictions.

(3) The Reviewer insists on his previous statement that a treatment which involves the anharmonic interaction would be a better (because simpler) explanation for the observed spectra.

Point (3):

Concerning point (3), we emphasize again that our treatment is solely based on the potential energy Hessian matrix calculated at the equilibrium position of a glass obtained by quenching in MD simulations, as done in all the papers quoted by the Reviewer. In all numerical treatments the density of states, which is the main subject of the discussion, is obtained from the eigenvalues of this (by definition harmonic) Hessian matrix. Our theory, along with all the numerical results derived from Hessian diagonalization, is inherently harmonic. It is important to note that the inherent anharmonicity, which is certainly present in both real and simulated glass dynamics at finite temperatures, is by definition not taken into consideration in these types of studies.

Point (2):

Regarding point (2), we would like to reiterate that the one-sided harmonic potentials exhibit a discontinuous second derivative at the truncation point, resulting in discontinuous Hessian elements ($m=1$ in our notation). Due to this characteristic, we opted not to incorporate this potential in a theory where the Hessian plays a pivotal role.

Moreover we would like to comment on the results of Lerner et al, [38] which the Reviewer quotes as evidence against the generality of our treatment. In this study the authors used 3 different potentials, namely a Kob-Anderson binary Lennard-Jones potential (KABLJ), a 3D repulsive inverse-power-law potential (3DIPL) and a one-sided harmonic potential (HARM). The first two had been truncated by $m=2$ tapering and show a density of states (DOS) scaling as $g(\omega)\sim\omega^4$, in agreement to our theory for rather low parental temperature. The Reviewer claims that the HARM data would also scale as $g(\omega)\sim\omega^4$. At close inspection we find that, instead of 4.0, the exponent is 3.56 ± 0.2 . In an Erratum to Ref. [38], PRL 119, 099901 (2017) Lerner et al. concede that the parental temperature for this system was higher than that of the other ones. It has been shown in Ref. [44] that by increasing the parental temperature the exponent of the DOS is continuously decreased. In our theory a high parental temperature corresponds to a marginal situation with exponent 2. So, even if we have concerns considering the one-sided harmonic potentials, the results of [38] do not contradict our predictions.

Conclusion

In conclusion, we would like to emphasize that our treatment does not primarily focus on explaining in details the numerical intricacies of the observed scaling exponents. Instead, our main objective is to illustrate their dependence on the potential tapering through the analysis of the statistics of local frozen-in stresses. Furthermore, we highlight the significance of the parental temperature, influencing the formation of structures near marginal stability or deviating from it.

To better underscore the absence of role of anharmonicity and elucidate the rationale behind our decision not to investigate HARM-like potentials, we have incorporated the following two paragraphs into the main text:

At the end of “Resulting scenario..”:

The Hessian, however, on which our analytical theory is based, and from which the DOS is obtained by diagonalization, plays a pivotal role. Therefore we decide not to consider the cases with $m < 2$, where some matrix elements of the Hessian are discontinuous at the truncation point.

At the end of the conclusions:

At the end we would like to emphasize that the present theory, along with the previous numerical results, obtained by diagonalizing the Hessian matrix (e.g. [38–45]), is inherently harmonic. Anharmonic effects, which certainly contribute to experimentally measured spectra (e.g [4–9]) and spectra obtained by evaluating correlation functions, using molecular-dynamics simulations (e.g. [25–28]), are not taken into account. The harmonic spectra pertain to zero temperature, while the anharmonic effects vanish in this limit. However, the experimentally measured non-phononic spectra in the THz range (boson peak) are reportedly dominated by the harmonic interaction [10,68].

REVIEWERS' COMMENTS

Reviewer #1 (Remarks to the Author):

I think the authors addressed the third reviewer's concern in somehow good way, even though I also believe anharmonicity should be very important in both simulation and experimental glasses and liquids (as I raised in the first round). The harmonic approximation provides some simple answers to understand the glass vibrations.

Reviewer #2 (Remarks to the Author):

I have carefully reviewed the comments made by the third reviewer, the author's rebuttal letter, and their updated manuscript. The third reviewer raised two issues. The first issue was about the exclusion of one-sided harmonic interactions in the author's manuscript. In my opinion, this is justified because the harmonic interactions between soft repulsive particles can cause technical issues, leading to some elements of Hessian being ill-defined at zero temperature. The second issue raised by the third reviewer is about anharmonic interactions that are not considered in the author's theory. However, at zero temperature, a strictly harmonic approach through Hessian is still a good approximation, even though authentic glasses and some simulated ones at finite temperatures always have anharmonic interactions due to high-order terms beyond the harmonic expansion of the interaction potential. Therefore, I believe that the authors' handling and argument of the above two issues are appropriate, and I continue to support the acceptance of their paper.

Reviewer #3 (Remarks to the Author):

The authors reexamined the simulation data in Ref. [38] and write in the reply, "At close inspection we find that, instead of 4.0, the exponent is 3.56 ± 0.2 ." Is such a small difference reliable? In the original paper, the authors have concluded that the ω^4 scaling is independent of the potential.

Based on the reply, I conclude that the scaling behavior in HARM systems cannot be explained by the current theory. The origin of the scaling in the HARM model remains an open question. Other than this, the results are reliable and valuable. It is a pity that the authors do not wish to make an effort to incorporate such an important model in their theory (at least in the current work). Although the matrix elements of the Hessian can be discontinuous in the HARM model, the DOS of the HARM model can be nevertheless consistently measured in simulations. Thus it should be possible to treat the HARM model within a theoretical framework based on continuous Hessian elements, with some approximations or additional theoretical considerations.

Since the above limitation of the current theory has been specified, I think that the current manuscript can be published.